



# Implementation of additional spectral wave field exchanges in a 3D wave-current coupled WAVEWATCH-III (version 6.07) - CROCO (version 1.2) configuration and assessment of their implications for macro-tidal coastal hydrodynamics

Gaetano Porcile[1], Anne-Claire Bennis[1], Martial Boutet[1], Sophie Le Bot[2], Franck Dumas[3], and Swen Jullien[4]

[1]Normandie Univ., UNICAEN, UNIROUEN, CNRS, M2C, 14000 Caen, France
[2]Normandie Univ., UNIROUEN, UNICAEN, CNRS, M2C, 76000 Rouen, France
[3]Shom, Brest, France
[4]Ifremer, Univ. Brest, CNRS, IRD, LOPS, IUEM, F-29280, Plouzané, France

**Correspondence:** Gaetano Porcile (gaetano.porcile@edu.unige.it)

**Abstract.** An advanced coupling between a three-dimensional ocean circulation model (CROCO) and a spectral wave model (WAVEWATCH-III) is presented to better represent wave-current interactions in coastal areas. In the previous implementation of the coupled interface between these two models, some of the wave-induced terms in the ocean dynamic equations were computed from their monochromatic approximations (e.g., Stokes drift, Bernoulli head, near-bottom wave orbital velocity, wave-to-ocean energy flux). In the present study the exchanges of these fields computed from the spectral wave model are implemented and evaluated. A set of numerical experiments for a coastal configuration of the circulation near the Bay of Somme (France) is designed. The impact of the spectral versus monochromatic computation of wave-induced terms significantly affects the hydrodynamics at coastal scale in the case of storm waves and winds opposed to tidal flows, reducing the wave-induced deceleration of the vertical profile of tidal currents. This new implementation provides current magnitudes closer to measurements than those predicted using their monochromatic formulations, particularly at the free surface. The spectral surface Stokes drift and the near-bottom wave orbital velocity are found to be the most impacting spectral fields, respectively increasing advection towards the free surface and shifting the profile close to the seabed. In the particular case of the Bay of Somme, the approximation of these spectral terms with their monochromatic counterparts ultimately results in an underestimation of ocean surface currents. Our model developments thus provide a better description of the competing effects of tides, winds, and waves on the circulation of coastal seas with implications to the study of air-sea interactions and sediment transport processes.

## 1 Introduction

The majority of the world's population live in coastal environments and the demographic predictions indicate that it will further increase in the future (Rao et al., 2008; Ioc-Unesco and FAO, 2011). The study of coastal systems therefore becomes



a priority to better conciliate nature preservation and human activities, particularly in the current context of climate change.
Coastal and estuarine dynamics are driven by several forcings, notably tides, wind- and wave-induced currents and setup. The
hydrodynamics in these areas induces sediment transport of fine to coarse materials, which shapes the bottom morphology, and
thus in turn impacts the hydrodynamics. The various components of the coastal system are thus strongly coupled and encompass
scales ranging from several kilometers to less than a meter. To understand this complex dynamics, numerical modelers have
worked during the last decades at coupling the different components of the coastal system.

The study of complex nearshore processes primarily driven by ocean waves is currently performed by combining wave
models (spectral, monochromatic or wave-resolved) with hydrodynamic models. To couple these models, wave forcing terms
are usually added to the momentum equations, while current and water level forcing terms are added to the wave action equation
when using spectral wave models. For two-dimensional horizontal (2DH) cases, equations derived by Phillips (1977) based on
the pioneer works of Longuet-Higgins and Stewart (1962, 1964) using the wave radiation stress concept have been successfully
implemented in numerical models to simulate wave-induced dynamics. These equations considering total mass transport were
adapted by Smith (2006) who separated the mass transport due to waves from that caused by the mean circulation and were
used, for instance, to simulate in a 2DH framework the rip current dynamics and very low frequency motions at Biscarosse
beach (France, Bruneau et al. (2011)). To improve the understanding of nearshore dynamics, notably to reproduce the undertow
vertical structure, three-dimensional modelling of the wave-induced flow was requested. Two types of theories have been
developed: the first considers the mean flow (e.g. McWilliams et al., 2004; Ardhuin et al., 2008) while the second is based on
the total current (e.g. Aiki and Greatbatch, 2012, 2013; Nguyen et al., 2021). The final sets of equations obtained from these
theories were successfully implemented in many hydrodynamic models for coastal applications (e.g. Uchiyama et al., 2010;
Bennis et al., 2011; Kumar et al., 2011, 2012; Michaud et al., 2012; Moghimi et al., 2013; Bennis et al., 2014) and global (e.g.
Couvelard et al., 2020), with simplifications in some cases.

In the context of the development of the Coastal and Regional Ocean COmmunity model (CROCO, https://www.croco-
ocean.org), the present paper contributes at investigating the sensitivity of coastal hydrodynamics to the implementation of
wave-induced terms. CROCO is a new oceanic modeling system built upon ROMS-AGRIF (Shchepetkin and McWilliams,
2004; Penven et al., 2006; Debreu et al., 2012), and gradually including new features such as a non-hydrostatic kernel (Hilt
et al., 2020), and coupling with several modules and models (atmosphere, surface waves, marine sediments and biogeochem-
istry). Wave-current coupling in CROCO was initiated by Marchesiello et al. (2015) to simulate the nearshore dynamics by
implementing the vortex force formalism of McWilliams et al. (2004) according to Uchiyama et al. (2010). In this formal-
ism, the mean flow, represented by the mean Eulerian velocity, is solved and the wave effects are modeled thanks to wave
forcing terms, which include non-dissipative and dissipative processes. A monochromatic wave model (WKB) was directly
implemented in CROCO following Uchiyama et al. (2010) to compute the wave dynamics. Marchesiello et al. (2015) have
tested this implementation against classical test cases (e.g. planar beach, barred beach, rip currents) and their results are sim-
ilar to the former ones found in Uchiyama et al. (2010), validating the implementation. They also successfully modelled the
real case of the Biscarosse beach (France), with a good representation of the rip current dynamics and the expected strong
cross-shore velocities. In parallel, developments were carried out to implement a coupled interface for the air-sea exchanges



including those between waves and currents by means of the OASIS-MCT coupler (Valcke et al., 2015). The interface provides a non-intrusive and flexible framework to couple with any other model with a similar interface. The 3-way coupling (e.g. ocean-wave-atmosphere) was tested against observations (in-situ and satellite) for the case of tropical cyclone Bejisa (2013–2014) by Pianezze et al. (2018), and a 3-way coupling tutorial configuration of the Benguela region (South Africa) is also available for the user community via the CROCO documentation and tutorials (https://doi.org/10.5281/zenodo.7400922).

In these former works, the exchanges of wave terms through the OASIS-MCT coupler only included significant wave height, mean or peak direction and frequency. The wave-induced terms were then computed using a monochromatic approximation following the implementation made with the WKB model. However, in cases where the sea state is more complex than a single close-to-monochromatic wave system (e.g. multi-modal or spread spectra), the wave-induced terms computed from the full spectrum (which are provided by spectral wave models) may be significantly different from their monochromatic

approximation. This paper thus presents the implementation of the spectral wave-induced terms in CROCO and evaluate their added-value in a coastal simulation of the Bay of Somme (France).

In the following section the study site is introduced along with the observational data used to setup and validate the model. Section 3 is devoted to the description of the methodology. The spectral wave model is introduced along with the hydrodynamic model. The implementation of additional spectral wave-induced terms in the hydrodynamic computation is detailed. The

different coupling procedures are described and the performed numerical experiments are summarised. Section 4 presents the numerical results. Modelled waves and currents are validated against in-situ measurements. Then, the sensitivity of structure of the modelled nearshore currents to the newly-added spectral vs monochromatic wave-induced terms is provided considering contrasting events in terms of wind forcing relative to tidal currents. Section 5 is devoted to the discussions and conclusions.

## 2 Application site and data

The application site, Bay of Somme (hereafter named BoS), is located in the eastern part of the English Channel near Dieppe-Le Tréport (Seine-Maritime, France). BoS is the tidal inlet shown in Fig. 1.

The coastal dynamics off the bay is mainly influenced by marine (waves and tide), meteorological (wind and sea-level pressure) and fluvial (Somme's river) effects. Semi-diurnal tide is the main hydrodynamic forcing with a macro-tidal range of 8.5 m for an average spring tide and reaching 10.55 m for exceptional tides (SHOM, 2020). Tidal currents are bi-directional,

oriented off the BoS to the east-northeast and west-southwest respectively during flood and ebb tides. Inside BoS, they flow to the east turning to southeast and to the west turning north-west for flood and ebb, respectively. Tidal asymmetry is present (SHOM, 2020), with an ebb flow (surface velocity going up to 0.95 m/s and 2.09 m/s off BoS and at its entrance, respectively) which is weaker than the flood (surface velocity going up to 1.2 m/s and 2.5 m/s off BoS and at its entrance, respectively). Ocean waves also affect hydrodynamics and are responsible for sedimentary movements offshore the bay (Ferret et al., 2010)

and in the nearshore (Michel et al., 2017; Turki et al., 2021). The climatological significant wave height and period are 2 m and 7 s, respectively (Ferret et al., 2010). This indicates a not fully developed wind sea. The energetic swells are generated on limited fetches with a maximum length of 400 km.



Strong macro-tidal currents and energetic storms acting on limited fetches less than 100 km long for 2/3 of the time and only exceeding 500 km in length for 1/20 of the time (Ferret, 2011) induce sediment transport, resulting in a variety of bedforms, which in turn locally control the hydrodynamics. Offshore BoS, the seabed is composed of a mixture of coarse to fine sands moulded with tidal sandbanks covered with very large sand dunes (Ferret et al., 2010). Within BoS, the seabed sediment displays a fining gradient from fine sands in the external part of the bay to muddy fine sands in some of its internal parts (Michel et al., 2017). Widespread dynamic morphological features are present, such as small to large dunes, tidal channels and sand bars (Michel et al., 2017). As most of estuaries and bays, BoS is concerned with an infilling sedimentary process in relation with marine sediment contribution and the erosion of the front submarine part of the bay sedimentary prism (Verger, 2005; Michel et al., 2017).

In-situ data used for the validation step (wave characteristics, flow velocity magnitude and direction as well as water levels) were recorded off BoS over a submarine sand dune field located in the southwest region of the study area. These data, acquired using an Acoustic Doppler Current Profiler (ADCP) and an AquaDopp (AQDP) during a field campaign in summer 2008 (MOSAG08 survey), have been previously analysed by Ferret (2011). Only data of ADCP C3 (1 MH$_z$; SonTek©Instruments) and AQDP B (2 MH$_z$; Nortek©) from 22 July to 6 August 2008 were used to validate our numerical application, because wave conditions were the most energetic at C3 location (significant wave height reaching 2 m). The geographical positioIndeed wave-induced currents play a crucial role in determining the coastal circulation patterns in the area of interest. They are properly modelled in Delft3D and lead to the convergence and to the offshore oriented rip when the rotation of the incoming waves is clockwise such as during the modelled Melor typhoon. (@GAETANO: aggiungere una linea tratteggiata in Figura 3 dove mostrare dove le onde iniziano a frangere?)ns of C3 and B mooring stations were 50°09,372′N/1°17,026′E and 50°08,851′N/1°17,521′E (Fig. 1, red marks). Both profilers were immersed at a depth of 13.5 m (Lowest Astronomical Tide chart datum). Measurements are the result of a 1 min average operated every 5 min and 12 min respectively for C3 and B. These measurements were recorded at 1 meter above the seabed for B and over the whole water column for C3, with a 0.4 m bin resolution and a first measure taken at 2 meters above the bottom (mab). At these locations, a median sediment diameter equal to 200 $\mu$m was observed by Ferret (2011).

## 3 Methodology

This section presents the implemented methodology. First, the spectral wave and hydrodynamic models are introduced. Then, the new implementations for the modelling of wave-current interactions are described in detail. Lastly, the employed coupling procedures and performed numerical experiments are presented.

### 3.1 Spectral wave model

WAVEWATCH-III (Tolman, 2016) is a community wave modelling framework that includes the latest scientific advances in the field of wind-wave modelling and dynamics. The WAVEWATCH-III third-generation wave model was developed at the US



National Centers for Environmental Prediction (NOAA/NCEP). In this study the version 6.07 is employed (WAVEWATCH-
III©, 2019).

WAVEWATCH-III computes surface gravity wave propagation solving the discrete phase spectral action density balance
equation for wavenumber-direction spectra and accounts for the main physical processes influencing the propagation of waves
as sinks or sources of wave energy:

$$\frac{\partial}{\partial t}\left(\frac{E}{\omega}\right) + \frac{\partial}{\partial x}\left(c_{gx}\frac{E}{\omega}\right) + \frac{\partial}{\partial y}\left(c_{gy}\frac{E}{\omega}\right) + \frac{\partial}{\partial \theta}\left(c_\theta\frac{E}{\omega}\right) + \frac{\partial}{\partial \omega}\left(c_\omega\frac{E}{\omega}\right) = \frac{S}{\omega}, \qquad (1)$$

where $E = H_s^2/16$ is the wave energy proportional to the square of the significant wave height $H_s$, $\omega$ is the wave frequency,
$(cg_x, cg_y)$ are the group wave velocity components, $\theta$ is the wave direction, $c_\theta$ represents the wave propagation velocity in
the $\theta$-space characterising the shifting of the relative frequency due to variations in water depth and current, while $c_\omega$ is the
propagation velocity in the $\omega$-space that describes the depth- and current-induced wave refraction. The implicit assumption
of this equation is that properties of medium (water depth and current) as well as the wave field itself varies on temporal
and spatial scales that are much larger than the variation scales of a single wave. The third-order accurate numerical scheme
(Leonard, 1979) is presently used to describe wave propagation in combination with the total variance diminishing limiter
(Tolman, 2002).

Source terms are integrated in time using a dynamically adjusted time stepping algorithm, which concentrates computational
efforts in conditions with rapid spectral changes. In deep waters, the dominant source terms are wave growth due to wind action
$S_{in}$, nonlinear wave-wave interactions $S_{nl}$, and white-capping $S_{ds}$. Presently, nonlinear wave-wave interactions are modeled
using the discrete interaction approximation (Hasselmann et al., 1985) while input and dissipation source terms are based
on Ardhuin et al. (2010). Proceeding into shallow waters, additional source terms must be included as bottom friction $S_{bot}$
and depth-induced breaking $S_{db}$. In this study, the parameterization of bottom friction for sandy bottoms (Tolman, 1994) is
employed as later calibrated by Ardhuin et al. (2003) to field measurements (Zhang et al., 2009). Depth-induced breaking is
modeled following the formulation of Battjes and Janssen (1978). The model includes options for wetting and drying of grid
points.

### 3.2 Hydrodynamic model

CROCO (Coastal and Regional Ocean COmmunity model, https://doi.org/10.5281/zenodo.7415343) is a new oceanic mod-
elling system built upon the ROMS-AGRIF model (Shchepetkin and McWilliams, 2004; Penven et al., 2006; Debreu et al.,
2012). It solves finite-difference approximations of the Reynolds-averaged Navier–Stokes Equations (RANS) on a horizontal
free-surface Arakawa C grid and vertical stretched terrain-following coordinates with a split-explicit time stepping algorithm.
CROCO has a flexible structure that allows to choose several numerical schemes and parameterizations. Here we use the model
with the hydrostatic and Boussinesq approximations, the WENO5 scheme (Acker et al., 2016) for horizontal and vertical ad-
vection, the generic length scale scheme for vertical mixing based on the k-$\epsilon$ turbulent closure (Jones and Launder, 1972;
Umlauf and Burchard, 2005; Warner et al., 2005), and the parameterization of subgrid-scale processes in the bottom boundary



layer considering the combined wave-current drag as in Soulsby and Clarke (2005). Momentum, scalar advection, and diffusive processes are represented using transport equations.

### 3.3 Wave-current interactions and new implementation of spectral wave-induced terms

For the phase-averaged wave-current interactions, the following equations are solved in Eulerian framework and Cartesian
coordinates (Uchiyama et al., 2010; Marchesiello et al., 2015) using hydrostatic, Boussinesq, and incompressible assumptions:

$$
\begin{cases}
\nabla \cdot \mathbf{v_L} = 0, \\
\dfrac{\partial u}{\partial t} + \nabla \cdot (\mathbf{v_L} u) - f v_L = -\dfrac{\partial \phi^c}{\partial x} + \left( u_s \dfrac{\partial u}{\partial x} + v_s \dfrac{\partial v}{\partial x} \right) + \mathcal{F}_u + \mathcal{D}_u + \mathcal{F}_u^w, \\
\dfrac{\partial v}{\partial t} + \nabla \cdot (\mathbf{v_L} v) + f u_L = -\dfrac{\partial \phi^c}{\partial y} + \left( u_s \dfrac{\partial u}{\partial y} + v_s \dfrac{\partial v}{\partial y} \right) + \mathcal{F}_v + \mathcal{D}_v + \mathcal{F}_v^w, \\
\dfrac{\partial \phi^c}{\partial z} + \dfrac{\rho g}{\rho_0} = \mathbf{v}_s \cdot \dfrac{\partial \mathbf{v}}{\partial z},
\end{cases}
\tag{2}
$$

where $\mathbf{v}_L = (u_L, v_L, w_L)$ is the phase-averaged Lagrangian velocity, $\mathbf{v} = (u, v, w)$ is the phase-averaged Eulerian velocity and $\mathbf{v}_s = (u_s, v_s, w_s)$ is the 3D Stokes velocity. The phase-averaged Lagrangian velocity is calculated such that $\mathbf{v}_L = \mathbf{v} + \mathbf{v}_s$. $\mathcal{D}_u$ and $\mathcal{D}_v$ are diffusive terms including wave-enhanced drag and mixing. $\mathcal{F}_u$ and $\mathcal{F}_v$ are forcing terms (in the present study we only
consider winds at the free-surface), while $\mathcal{F}_u^w$, $\mathcal{F}_v^w$ are wave-induced forcing terms (including bottom streaming and breaking acceleration). $\rho_0$ is the reference density, $g$ is the gravity acceleration and $f$ is the coriolis parameter. $\phi^c = \phi + \hat{\phi}$ is related to the fluid pressure where $\phi$ is the dynamic pressure calculated such that $\phi = \dfrac{P}{\rho_0}$ (with $P$ the total pressure) and $\hat{\phi}$ is the Bernoulli head due to waves. In the v1.1 of CROCO, the wave-induced terms in the ocean dynamic equations (Stokes drift, Bernouilli head, bottom wave orbital velocity, wave-to-ocean energy flux) are computed from their monochromatic approximations.
In the present study, we have implemented the exchanges of these fields computed from the spectral wave model. These implementations are now included in CROCO v1.2. The formulation of the wave-induced terms in these two CROCO versions are detailed below.

#### 3.3.1 Stokes drift

In CROCO v1.1, the Stokes velocity is calculated from the monochromatic formulation such that:

$$
\begin{aligned}
u_s &= \frac{A^2 \sigma}{2\sinh^2(kD)} \cosh(2k(z+h)) k_x, \\
v_s &= \frac{A^2 \sigma}{2\sinh^2(kD)} \cosh(2k(z+h)) k_y,
\end{aligned}
\tag{3}
$$


where $A$ is the wave amplitude, $\sigma$ is its intrinsic frequency, $\mathbf{k} = (k_x, k_y)$ is the wavenumber vector, $D$ is the mean depth, $h$ is the bathymetric depth and $z$ is the vertical coordinate. Due to the non-divergence of the Stokes velocity, the vertical component is obtained from the horizontal ones:

$$
w_s = -\int_{-h}^{z} \left( \frac{\partial u_s}{\partial x} + \frac{\partial v_s}{\partial y} \right) dz'.
\tag{4}
$$



As the Stokes drift velocity is known to be dependent on wave frequencies at the surface, we have implemented in the new CROCO v1.2 the use of the spectral formulation of the velocity at the free surface $\mathbf{v}_{ss} = (u_{ss}, v_{ss})$ computed by WAVEWATCH-III as follows:

$$(u_{ss}, v_{ss}) = \int \int \sigma \cosh(2kD) \frac{(k\cos(\theta_w), k\sin(\theta_w))}{\sinh^2(kD)} F(k, \theta_w) dk d\theta_w, \tag{5}$$

where $F(k, \theta)$ is the wavenumber-direction energy spectrum (with $\theta_w$ the mean wave direction). From these surface Stokes velocity components, the dispersion relationship ($\sigma^2 = gk\tanh(kD)$), and the expansion of the hyperbolic functions describing the vertical distribution of the wave field, it is possible to obtain the following formulation for the 3D Stokes velocity field through the water column:

$$(u_s(z_r), v_s(z_r)) = (u_{ss}, v_{ss}) \left[ \frac{1}{2k(1 + e^{-4kD})} \right.$$
$$\left. \frac{1}{z_{up} - z_{low}} \left[ \left( e^{2k(z_{up}+h-D)} - e^{-2k(z_{up}+h+D)} \right) - \left( e^{2k(z_{low}+h-D)} - e^{-2k(z_{low}+h+D)} \right) \right] \right], \tag{6}$$

where $z_r$, $z_{up}$ and $z_{low}$ represent respectively the vertical coordinate of the levels at RHO-points (located at the centre of the computational cells) and those of the surrounding PSI-points (located at the edge of the computational cells). The interested reader is referred to the CROCO manual (https://doi.org/10.5281/zenodo.7400922) for a comprehensive description of the staggered computational grids and the vertical terrain-following sigma-layering.

### 3.3.2 Bernoulli head

The wave-induced pressure called Bernoulli head ($\hat{\phi}$) is computed in CROCO v1.1 with the following monochromatic formulation:

$$\hat{\phi} = \frac{A^2 \sigma}{4k \sinh^2(kD)} \int_{-h}^{z} \frac{\partial^2 \mathbf{k} \cdot \mathbf{v}}{\partial z'^2} \sinh(2k(z - z')) dz'. \tag{7}$$

In the new v1.2, we have implemented the use of the spectral Bernoulli head computed by WAVEWATCH-III:

$$\hat{\phi} = g \int \int \frac{k}{\sinh(2kD)} F(k, \theta) dk d\theta. \tag{8}$$

### 3.3.3 Near-bottom wave orbital velocity

Ocean waves also produce changes in bottom friction due to the enhancement of bottom drag and mixing as well as streaming effects. The parameterization of Soulsby (1997) is used for modelling bottom stresses in the presence of waves,

$$\tau_{wc} = \tau_c \left( 1 + 1.2 \left( \frac{\tau_w}{\tau_w + \tau_c} \right)^{3.2} \right), \tag{9}$$

where current ($\tau_c$) and wave ($\tau_w$) related shear stresses are:

$$\tau_c = \frac{\kappa^2}{\ln^2(z/z_0)} |\mathbf{u}|^2, \tag{10}$$





with $z_0$ the bottom roughness length and $\kappa$ the Von-Karmàn constant,

$$\tau_w = \frac{\rho f_w u_w^2}{2},\tag{11}$$

with $f_w = 1.39 \left(\frac{u_w}{\sigma_p z_0}\right)^{-0.52}$ the wave friction factor. $u_w$ is the near-bottom wave orbital velocity, which is calculated in v1.1
using its monochromatic formulation:

$$u_w = \sigma_p \frac{H_s}{2\sinh(kD)},\tag{12}$$

where $H_s$ is the significant wave height and $\sigma_P$ is the peak wave frequency from the linear wave theory (Airy, 1845). In the
newly implemented v1.2, the spectral near-bottom wave orbital velocity computed by WAVEWATCH-III is used instead of its
monochromatic counterpart:

$$u_w = \sqrt{2}\left(2 \int \int \frac{\sigma^2}{\sinh^2(kD)} F(k,\theta) dk d\theta\right)^{1/2}.\tag{13}$$

### 3.3.4 Wave-to-ocean energy flux

In the wave-averaged momentum equations of the CROCO model, the acceleration induced by wave breaking enters as a body
force:

$$\mathcal{F}_{u,v}^w = \frac{\epsilon_b}{\rho\sigma} k_{x,y} f_b(z),\tag{14}$$

where $f_b(z)$ is a normalised vertical distribution function representing the vertical penetration of momentum, and $\epsilon_b$ is the
depth-integrated rate of wave energy dissipation due to wave breaking. The parameterization for $\epsilon_b$ is crucial for both the wave
and the circulation model to respectively compute wave dissipation and associated current acceleration. In CROCO v1.1, only
few formulations of depth-induced wave breaking were implemented and these differ from those available in WAVEWATCH-
III. In this study the formulation of Battjes and Janssen (1978) is employed as it is currently available in WAVEWATCH-III
and its implementation in CROCO is straightforward. Furthermore, the spectral rate of wave breaking dissipation computed by
WAVEWATCH-III accounts also for deep-water breaking due to white capping such that:

$$\epsilon_b = \int \int S_{ds}(k,\theta) + S_{db}(k,\theta) dk d\theta,\tag{15}$$

where $S_{ds}$ is the deep-water dissipation term which includes the wave energy dissipation due to white capping (WAVEWATCH-
III©, 2019), while $S_{db}$ is the shallow-water dissipation term representing the bathymetric wave breaking. The latter is computed
in WAVEWATCH-III following Battjes and Janssen (1978):

$$S_{db}(k,\theta) = -0.25\alpha Q_b f_m \frac{H_{max}}{E} F(k,\theta),\tag{16}$$

where $\alpha$ is a tunable parameter ($\alpha = 1$ in this study), $H_{max} = \gamma D$ is the maximum height a component in the random wave field
can reach without breaking, $\gamma$ is a constant derived from field and laboratory observations ($\gamma = 0.73$ in this study), and $f_m$ is



the mean wave frequency. $Q_b$ is the fraction of breaking waves in the random field evaluated in terms of the ratio of $H_{max}$ and $H_{rms}$, which is the root-mean-square wave height such that:

$$\frac{1 - Q_b}{-ln(Q_b)} = \left(\frac{H_{rms}}{H_{max}}\right)^2 . \tag{17}$$

In CROCO v1.2, we have implemented the formulation of Battjes and Janssen (1978) for $\epsilon_b$ to compute the rate of wave energy dissipation due to depth-induced breaking using mean wave parameters as well as the alternative exchange of the spectral wave energy dissipation as directly provided by WAVEWATCH-III, including deep-water breaking (whitecapping) dissipation.

### 3.3.5 Turbulent mixing

As anticipated in the introducing paragraph of this section, the computation of the vertical viscous and diffusion coefficients is based on the generic length scale parameterization (Umlauf and Burchard, 2005) and specifically on the k-$\epsilon$ turbulence closure scheme (Jones and Launder, 1972). Thus, the eddy viscosity of momentum and eddy diffusivity of temperature read

$$\nu_T = c_\mu (k_T^2/\epsilon_T) \ , \quad D_T = c_\mu' (k_T^2/\epsilon_T) \ , \tag{18}$$

where $c_\mu$ and $c_\mu'$ are coefficients determined according to the stability functions of Canuto et al. (2001), while turbulent energy $k_T$ and energy dissipation $\epsilon_T$ are obtained from the following transport equations

$$\frac{Dk_T}{Dt} = \frac{\partial}{\partial z}\left(\frac{\nu_T}{S_{ck}}\frac{\partial k_T}{\partial z}\right) + \nu_T\left[\left(\frac{\partial u}{\partial z}\right)^2 + \left(\frac{\partial v}{\partial z}\right)^2\right] - \frac{g}{\rho_0}\frac{\partial \rho}{\partial z}D_T \ , \tag{19}$$

$$\frac{D\epsilon_T}{Dt} = \frac{\partial}{\partial z}\left(\frac{\nu_T}{S_{c\epsilon}}\frac{\partial \epsilon_T}{\partial z}\right) + \frac{\epsilon_T}{k_T}\left\{\beta_1\nu_T\left[\left(\frac{\partial u}{\partial z}\right)^2 + \left(\frac{\partial v}{\partial z}\right)^2\right] - \beta_3\frac{g}{\rho_0}\frac{\partial \rho}{\partial z}D_T - \beta_2\epsilon_T\right\} \ , \tag{20}$$

and the set of coefficients identified by Warner et al. (2005) is adopted: $S_{ck} = 1$, $S_{c\epsilon} = 1.3$, $\beta_1 = 1.44$, $\beta_2 = 1.92$ , $\beta_3 = 1$.

Since the turbulence model does not resolve the viscous sublayer, the boundary conditions are applied in this constant stress layer where it is assumed that the turbulent energy production equals its dissipation (Wilcox et al., 1998) and

$$k_{Tb} = (u_b^*)^2/(c_\mu^0)^2 \ , \quad k_{Ts} = (u_s^*)^2/(c_\mu^0)^2 \ , \tag{21}$$

where $c_\mu^0$ is a stability coefficient based on experimental data for unstratified channel flows with a log layer solution, $u^*$ is the friction velocity and subscripts $b$ and $s$ refer to bottom and surface, respectively. To ensure numerical stability, boundary conditions for the turbulent energy are also applied in flux form and assuming local steady state no gradient conditions:

$$\left(\frac{\nu_T}{S_{ck}}\frac{\partial k_T}{\partial z}\right)_b = 0 \ , \left(\frac{\nu_T}{S_{ck}}\frac{\partial k_T}{\partial z}\right)_s = 0 \ . \tag{22}$$

Boundary conditions for turbulent energy dissipation follow similar reasoning and yield

$$\epsilon_{Tb} = (u_b^*)^3/\kappa z_b \ , \quad \epsilon_{Ts} = (u_s^*)^3/\kappa z_s \ . \tag{23}$$



Flux conditions are specified also for turbulent energy dissipation to prevent numerical instabilities as follows

$$\left(\frac{\nu_T}{S_{c\epsilon}}\frac{\partial \epsilon_T}{\partial z}\right)_b = -\frac{\nu_T}{S_{c\epsilon}}(c_m u^0)^3\frac{\sqrt{k_{Tb}}}{z_b^2} \quad,\quad \left(\frac{\nu_T}{S_{c\epsilon}}\frac{\partial \epsilon_T}{\partial z}\right)_s = +\frac{\nu_T}{S_{c\epsilon}}(c_m u^0)^3\frac{\sqrt{k_{Ts}}}{z_s^2} \quad. \tag{24}$$

Wave dissipation induces additional mixing of momentum in the water column (Agrawal et al., 1992). Two main sources of wave energy decay are presently included, namely wave breaking at the free-surface due to depth-induced dissipation and whitecapping as well as bottom friction due to the oscillatory wave motion in the bottom boundary layer. These two sources of
260 wave dissipation are accounted for in the turbulence model by assuming an energy cascade in which the wave energy decay is transferred to the turbulent kinetic energy (Walstra et al., 2001). Wave energy dissipation due to bottom friction is considered to produce turbulent kinetic energy by increasing the bed shear stress in the bottom boundary layer (Eq.(9)). In the case of wave breaking, an additional production of turbulent energy is also considered directly associated with the depth-integrated rate of wave energy dissipation due to wave breaking (Deigaard et al., 1986). Following Kumar et al. (2012), this additional mixing
is incorporated in the k-$\epsilon$ model by introducing source terms in both the turbulent kinetic energy equation and the turbulent kinetic energy dissipation equation. Turbulence due to injection of surface flux of kinetic energy is given as surface boundary conditions (Craig and Banner, 1994; Feddersen and Trowbridge, 2005):

$$\left(\frac{\nu_T}{S_{ck}}\frac{\partial k_T}{\partial z}\right)_s = \epsilon_w \quad, \tag{25}$$

where $\epsilon_w$ is the downward flux of kinetic energy due to wave breaking. The surface boundary condition for $\epsilon_T$ due to breaking
waves is (Carniel et al., 2009):

$$\left(\frac{\nu_T}{S_{c\epsilon}}\frac{\partial \epsilon_T}{\partial z}\right)_s = -\frac{S_{ck}}{S_{c\epsilon}}(c_\mu^0)^3\frac{3}{2}\sqrt{k_T}\frac{\mathcal{Y}}{\kappa(z_0-z_s)} + \frac{\nu_T}{S_{c\epsilon}}(c_\mu^0)^3\sqrt{k_T}k_T\frac{(z_0-z_s)}{\kappa} \quad, \tag{26}$$

where $\mathcal{Y}$ is the surface mixing length. In the case of breaking waves, the surface mixing length is provided using the closure model of Stacey (1999):

$$\mathcal{Y} = \epsilon_w z_0 \quad, \tag{27}$$

with $z_0 = \alpha_w Hrms$ and $\alpha_w = 0.5$. Only part of the wave energy dissipation ($\epsilon_b$) contributes to turbulence mixing. The contribution of wave energy dissipation as surface flux of kinetic energy is expressed through an empirical coefficient $c_{\epsilon w}$. Thus, the downward flux of kinetic energy due to wave breaking is

$$\epsilon_w = c_{\epsilon w}\epsilon_b \quad, \tag{28}$$

According to Jones and Monismith (2008), we assume that only 5 % of wave energy dissipation goes into the water column as
turbulent kinetic energy.

### 3.4 Coupling procedure

In this study, CROCO and WAVEWATCH-III are coupled as presented in Fig. 2. Instantaneous hydrodynamic fields are exchanged between both models every coupling time step ($\Delta t_{coupling} = 60s$) thanks to the OASIS coupler (Valcke et al., 2015),



according to a similar procedure than in Pianezze et al. (2018); Bennis et al. (2020, 2022). In CROCO v1.1, WAVEWATCH-III
provides only three mean wave parameters based on the integration of the wave spectrum (Fig. 2 left panel, blue labels): the
significant wave height ($H_S$), the mean wave period ($T_{0M1}$), and the mean wave direction ($\theta_w$), which are used by CROCO to
compute wave-induced terms in the hydrodynamic equations, including wave-averaged set-up/set-down, horizontal and verti-
cal vortex force, wave-induced pressure, wave-induced tracer diffusivity, non-conservative wave dissipation, non-conservative
wave accelerations for currents and wave-enhanced vertical mixing (Eq. (2)). In the new v1.2, we have implemented the ad-
ditional exchanges of the mean wave length (LM), near-bottom wave orbital velocity (UBR), magnitude and direction of the
surface Stokes drift (USS), Bernouilli head pressure (BHD), and wave to ocean energy flux (FOC) from WAVEWATCH-III
to CROCO (Fig. 2 right panel, red labels). Thus, the wave-induced terms are computed from the full spectrum in the wave
model instead of using their monochromatic approximations, allowing a wider range of applications. In two-way coupling,
in addition to the fields sent from WAVEWATCH-III to CROCO, CROCO provides the sea surface height and surface flow
velocity to WAVEWATCH-III, which are used in the wave model to compute depth- and current-induced wave refraction.

### 3.5 Numerical experiments

Each numerical experiment performed in this study considers a rectangular computational domain delimited by the points
$1.156°E, 50.316°N$ and $1.781°E, 50.083°N$, respectively at NW and SE corners (Fig. 1). The same computational grid is
adopted for both the wave and the circulation models in the present study, but the use of different grids is possible. The
horizontal mesh has a spatial resolution of 100 m and the bathymetry from HOMONIM (Shom, 2015) is interpolated over this
grid. 20 sigma layers are used for the vertical discretization. They are uniformly distributed between the seabed and the free
surface, resulting in a maximum layer thickness offshore of about 2 m and a minimum layer thickness onshore of about 5 cm.
The wave model uses 32 frequencies (0.04 - 0.7 s$^{-1}$) and 24 directions leading to a directional resolution of 15°. Bi-dimensional
(frequency and direction) wave spectra from the HOMERE database (Boudiere et al., 2013) are used to interpolate wave forcing
at the deep-water open boundaries. Along these open boundaries, tidal forcing is interpolated from the high-resolution (250 m)
PREVIMER atlas MANE focused on the east part of the English Channel, which includes 37 harmonic constituents (Pineau-
Guillou et al., 2014). A cold start is imposed as initial condition and no stratification due to temperature and salinity gradients
has been considered. Wind forcing as stresses at the free surface is provided by NCEP throughout hourly CFSR Reanalysis at
0.312° horizontal resolution (Saha et al., 2010; Center, 2014). No water and heat fluxes from the atmosphere are considered.
Hydrodynamic motions were not computed for depths smaller than 1 m while wave-induced forcing terms were activated for
depths greater than 2 m. The observed mean grain size of 200 $\mu$m (Ferret, 2011) is used to compute the effective Nikuradse
roughness length employed in the parameterizations of bottom friction (CROCO using Soulsby (1997) and WAVEWATCH-III
using Ardhuin et al. (2010), ST4 package).

To understand the impact on the macro-tidal hydrodynamics of BoS of each metocean forcing (tide, wind and waves) and
each additional spectral field exchanged compared to its monochromatic approximation, a total of 9 numerical experiments
have been performed. All simulations presented in this article are detailed in Table 1. A simulation forced by only tidal levels
and currents (CRX) has been initially performed to be used as baseline case, providing a pure tidal vertical current profile. To





assess the impact on the vertical profile of wind forcing alone, a second simulation with tidal and wind forcing (WND) has been performed but without accounting for waves. Waves are then considered in all subsequent simulations. The first simulation uses CROCO v1.1 configuration with computation of wave-induced terms from their monochromatic approximation. The five following simulations include each one a different additional spectral wave field exchange instead of its monochromatic approximation (mean wavelength LM, Bernoulli head pressure BHD, wave to ocean energy flux FOC, surface Stokes drift USS, and near-bottom wave orbital velocity UBR). Finally a last simulation, named 'full coupling' uses all the spectral wave-induced terms included in CROCO v1.2.

## 4   Results and discussions

To assess the performance of the CROCO - WAVEWATCH-III coupled model, we compared numerical results in terms of mean wave parameters, water level and current with in-situ ADCP and AQDP measurements ( red dots in Fig. 1) and with the HOMERE hindcast (Boudiere et al., 2013). Then, the impacts of a spectral wave forcing on current and water level are analysed in time and space for different met-ocean conditions, depths and bottom features.

### 4.1   Assessment of modelled waves

Sea states were simulated for a time period ranging from 1 to 6 August 2008. During this period, an energetic wind event (fresh to strong breeze with moderate to large waves) occurred on 2 August with a wind velocity magnitude at 10-m height above the sea reaching $10.5 \, \mathrm{m.s^{-1}}$ between 5 p.m. and 7 p.m. The root mean square significant wave height ($H_{rms}$) computed by WAVEWATCH-III turns out to be smaller than the observed one, with a maximum value around $1.05 \, \mathrm{m}$ instead of the observed $1.40 \, \mathrm{m}$, leading to an underestimation of about 25% by the model. By contrast, simulated $H_{rms}$, mean wave period ($T_{0M1}$), and direction ($\theta_w$) are close to those predicted by the HOMERE hindcast. For the entire time series, the correlation coefficient $r^2$ is 0.86 while RMSE is $0.23 \, \mathrm{m}$. This partly validates our numerical configuration. Since the HOMERE hindcast is alike in wind forcing, physical parameterizations, and computational grid size, it is consistent to obtain similar results. Moreover, this hindcast was used to force the spectral wave model at these boundaries. Slight differences between the two model predictions are thus supposedly due to the higher spatial resolution employed in the present study (100 m), which results in a better description of wave shoaling and refraction.

Bathymetry-controlled wave refraction processes are observed (e.g., Longuet-Higgins, 1956). This refraction causes a rotation of wave trains that become parallel to the bathymetric contours. The wavenumber velocity, which points in the direction of wave propagation, becomes perpendicular to the bathymetric contours, as shown in Fig. 4. Moreover, as the bathymetry-induced refraction generates a convergence (or divergence) of the wavenumber vectors, the wave amplitude is modified such that it increases (or decreases) for a convergent (divergent) motion. Otherwise, the waves are deviated from their direction of propagation by the surface currents (e.g., Ardhuin et al., 2012; Bennis et al., 2020). In the study area, the flood flow is oriented towards the east-northeast while the ebb flow goes to the west-southwest ( Fig. 6). Due to the tide asymmetry often observed in the English channel, the ebb current is less intense than the flood current with respectively a mean value of $0.45 \, \mathrm{m.s^{-1}}$ and



Figure 9 shows the comparisons of model predictions with the ADCP measurements in terms of current magnitude and direction at 1 meter below the free-surface (1mbfs). The modelled current magnitude is correctly predicted during most of the phases of the tidal cycle, particularly at tidal current reversal and especially during ebb phases. Also current directions are fairly replicated. The phase shift between measured and modelled currents is present also at the free-surface when comparing modelled currents with the ADCP measurements. It is worth noting that peak flood velocities delay is not coincident with passing storms, thus suggesting that winds and waves are not responsible for the phase shift and adding up to the likelihood that this is due to the presence of sub-grid scale bedforms. Results obtained by the standalone CROCO model forced only with tides (CRX) are modified by wind stresses (WND), resulting in an acceleration and a deceleration of surface currents respectively at tidal floods and ebbs due to winds blowing towards the east-northeast. This wind-driven modulation is pronounced during storms. Wave effects on currents tend to smooth this modulation (2WC) by means of wave-current interaction mechanisms, as explained in the following section.

Tests of model results (2WC) against in-situ measurements through the water column in terms of eastward (Fig. 10) and northward (Fig. 11) current velocity components at the ADCP location further validate our modelling. Overall, the fair matching between measured and modelled current velocity components during the different phases of the tidal cycle and within the entire water column proves the reliability of the hydrodynamic model. However, an overestimation (underestimation) of the northward tidal current velocity computed by the model is observed during ebb (flood) peaks, particularly around the low tide slack water event, from the surface to the bottom.

## 4.3 Assessment of vertical current profiles for contrasting events

To assess the importance of exchanging spectral wave fields instead of computing them from monochromatic approximations, we compared measured vertical profiles of currents extracted at noteworthy time instants within various simulations listed in Table 1. Figure 12 shows two vertical current profiles associated with storm waves and calm sea states. Measured (squares) and modelled (continuous lines) time-series of current velocities (red) and root-mean-square wave heights (blue) are compared in the top panel (A). Middle panel (B) analogously compares current and wave directions. Vertical profiles were extracted at two time instants characterised by the occurrence of waves coming from west with $H_{rms}$ respectively larger and smaller than 1 m. In the case of storm waves (bottom-left panel C), wind-driven stresses are shown to modify the logarithmic tidal velocity profile accelerating the current towards the free-surface and decelerating it towards the bottom, as reported in former studies (e.g., Davies and Lawrence, 1995). However, this velocity increase seems too strong in view of the data with an overestimate of about 5 cm/s at the surface, which represents about 18.5 % of the measured surface velocity. Wave-current coupling considering terms computed with their monochromatic approximations (CROCO v1.1, 2WC) produced a realistic surface flow which fits now very well to the measurements. The smoothing of the wind-induced profile is due to waves because of the wave-current interaction mechanism described in Groeneweg and Klopman (1998), which is activated by waves moving in a direction similar to that of the current ( Fig. 12 C). The adding of the full wave spectral terms (CROCO v1.2) results in a light smoothing of the vertical profile throughout the entire water column, but the overall impact on model predictions is minor ( Fig. 12, 2WF).





In the case of calm sea states (bottom-right panel D), wind and wave effects are negligible and do not significantly modify the logarithmic tidal velocity profile.

Figure 13 shows two vertical current profiles associated with $\sim 10\ m/s$ winds. Measured (squares) and modelled (continuous lines) time-series of current velocities (red) and wind speeds (blue) are compared in the top panel (A). Middle panel (B) analogously compares current and wind directions. Note that modelled wind speeds and directions from the CFSR global reanalysis data set are tested against the records of a Met Office mooring station in the English Channel just offshore the study area $(0.0°E, 50.4°N)$ from the Copernicus Marine In-Situ Near Real Time Observations of the Atlantic Iberian Biscay Irish Ocean (Copernicus Marine Service, 2021). Vertical profiles were extracted at two time instants characterised by the occurrence of winds coming from west and contrasting ebb flows and favouring flood flows. In the case of winds blowing in a direction which is opposite to that of tidal currents (bottom-left panel C), wind-driven stresses are shown to modify the logarithmic tidal velocity profile (black curve) by slowing the flow over a depth of 10 m due to wind resistance of the flow, which results in a reduction in the surface velocity magnitude of about 8 cm/s (or 25 %) of the pure tidal velocity. CROCO v1.2 coupling results in a current profile in-between the logarithmic tidal profile and that of CROCO v1.1. It is closer to that of the simulation only influenced by wind stresses with lower decrease of the close-to-surface velocity, and slightly increase of close-to-bottom velocity. The profile modelled with CROCO v1.2, while still showing significant biases with observations, is however the best fitting with measurements throughout the entire water column compared to other simulations ( Fig. 13, 2WF). Waves accelerate the wind-induced current of about 4 cm/s at the surface due to an angle between wave and current directions of propagation slightly larger than 90°. Indeed, the wave energy spectrum at this time shows a sea state with two peak frequencies (around 0.15 and 0.25 Hz) and as a result the use of spectral forcing terms improves the accuracy of the results ( Fig. 13; RMSE $\sim 1.8\ cm/s$, $r^2 = 0.99$). Differences between CROCO v1.1 and v1.2 are mainly related to the near-bed wave orbital velocity, which is increased by a factor of about 1.5 at this time (Fig. 17), leading to a reduction in the bottom stress (Eq.(9)) due to the weak value of the near-bottom flow velocity ( 20 cm/s). This reduces the flow intensity across the entire water column, as described by Bennis et al. (2020, 2022). In the case of winds blowing in the same direction of tidal currents (bottom-right panel D), winds and waves affect only slightly the tidal logarithmic profile, leading to less significant differences. Indeed, the wind forcing magnitude at 14.30 and at 23.00 is similar, but its impact on current is minor at 23.00 as the instantaneous surface velocity magnitude is more than twice as high as at 14.30 ( 60 cm/s versus 25 cm/s). As in the previous case (Fig. 12), wind stresses accelerate the tidal profile at the free-surface since wind is following the current, while waves tend to smooth it.

The velocity magnitude on the water column is reduced by the use of wave spectral forcing terms because of the increase in the near-bed orbital velocity (about 2 cm/s) that causes an enhancement of the bottom stress. It is also worth noting that, in both situations, maximum current magnitude in the middle of the water column are not captured by the model. Better fits are obtained near the surface than in the middle of the water column. This can be explained by the modelling of turbulent mixing. At 14.30, it appears that vertical profiles are smoother then observations, making think to a too high vertical viscosity coefficient due to RANS modelling. At 23.00, many breakings occur due to opposite wave and current directions of propagation and also to a wind velocity causing whitecaps. The wave-induced turbulence is transmitted thanks to a surface term (Eq. 25-26). This could be the origin of discrepancies since the mixing is not propagated in the water column, but just located at the





surface. Otherwise, at 14.30 and at 23.00, breakings make difficult the measurements due to the flow aeration. Thus, a part of the differences can also be caused by the measurement techniques and site conditions. Overall, stronger effects due to the newly exchanged spectral terms are predicted during storm conditions when incoming sea-state spectra are likely broad or
multi-modal.

## 4.4   Temporal sensitivity to spectral vs monochromatic wave-induced fields

This section is devoted to the assessment of the impact of spectral versus monochromatic computation of each newly exchanged wave-induced term on the hydrodynamics. Six simulations were performed adding only the exchange of the spectral fields one-by-one through the coupling interface and then comparing the results in terms of vertical current profiles. Figure 14 shows
these comparisons at time instants when the tidal current is affected by wind forcing ( also Fig. 13). The profiles predicted by the simulations including only the effect of the spectral mean wavelength (2WC+LM) or the effect of the spectral Bernoulli head (2WC+BHD) match the results of CROCO v1.1 (2WC), indicating that considering these two spectral quantities rather than their monochromatic counterparts does not significantly affect the local hydrodynamics. It is worth noting that the storm conditions presently analysed are mild if compared with major events and that in the case of more energetic conditions the
spectral Bernoulli head is thought to have implications on the surface flow field, especially in the case of complex sea-state spectra. Simulations including only the spectral wave energy dissipation due to breaking (2WC+FOC), or the Stokes drift (2WC+USS), or the near-bottom orbital velocity (2WC+UBR) present interesting deviations from that of CROCO v1.1 (2WC). The main effect is caused by the use of spectral near-bed velocity which induces a decrease of the bottom shear stress, leading to an increase of the velocity magnitude of 1/1.5 cm/s across the entire water column. The adding of the spectral Stokes drift at
the surface and its distribution over the depth according to Eq. (6) slightly changes the vertical shape of the current due to vortex force which redistributes the momentum. The velocity magnitude is also altered by the Stokes drift contribution to the vertical advection. Differently, wave-to-ocean energy flux in its spectral form only affects the first 5 m below the sea level. Indeed, this flux represents the wave breaking contribution to the circulation. As it is used for computing the breaking acceleration force and surface boundary condition for mixing, it reduces the flow velocity near the surface.
These newly exchanged spectral terms correct the solution predicted by coupling using their monochromatic counterparts when the sea-state spectrum is far from having most of the energy concentrated in one frequency. Figure 15 shows the spectra associated with vertical profiles of Fig. 13 and 14. As expected, the two-peak spectrum (Fig. 15 A, C) corresponds to the vertical profile when wind forcing opposes tidal current (left panel of Fig. 14), which is strongly affected by the spectral versus monochromatic computation of wave-induced terms. Conversely, the one-peak spectrum (Fig. 15 B, D) corresponds to the
vertical profile when the contributions of the spectral terms are minor (right panel of Fig. 14), as expected.

Differences between the spectral and monochromatic Stokes drifts computed at the ADCP location are shown in Fig. 16 during the entire simulation time. It can be seen that the free-surface (< 0.1 m/s) and depth-averaged (< 0.02 m/s) values of the Stokes drift are mainly governed by the occurrence of storms, with higher and longer waves associated with larger Stokes velocities (∼ 0.09 m/s). The surface Stokes velocity is 2 to 6 times stronger than its barotropic counterpart (2WC and 2WF
cases), showing a modulation in time of the vertical shape of the Stokes velocity with the largest changes for strong winds





and high waves. The magnitude of the Stokes drift computed by the spectral wave model (2WC, red and magenta lines in Fig. 16) turns out to be smaller than its monochromatic counterpart (2WF, blue and cyan lines in Fig. 16) the 2 August 2008 at 2.30 pm, while spectral and monochromatic values are higher than monochromatic ones the 2 August 2008 at 11 pm. Despite spectral and monochromatic values remain close (only about 1 cm/s differences), the impact of the spectral Stokes velocity at
the surface on the overall current profile is not negligible, especially at 14.30 as shown in Fig. 14.

Differently from the Stokes drift, the near-bottom wave orbital velocity is strongly modulated by the time evolution of the macro-tidal sea level range ( Fig. 17). The near-bed wave orbital velocity is increased during low tide, while a reduction in its intensity is observed during high tide (2WC and 2WF cases). This is primarily due to the fact that in shallower waters the action of waves close to the seabed is enhanced, especially in macro-tidal settings where water depths are of the same order
of tidal ranges as at the ADCP location. The modulation according to the tidal phase is more intense for the spectral velocity with an amplification up to 30 %. Indeed, the spectral velocity takes into account the doppler effect generated by tide, while the monochromatic velocity does not consider this effect as it makes use of a peak wave frequency from the linear theory of Airy (1845).

### 4.5  Spatial sensitivity to spectral vs monochromatic wave-induced fields

Differences between the Stokes drift predicted by the spectral model when coupled within CROCO v1.2 framework (2WF) and its monochromatic approximation computed when coupled within CROCO v1.1 framework (2WC) are shown for two time instants in the top and bottom panels of Fig. 18. The spectral Stokes drift is accelerated for depth smaller than 5 m, which corresponds to the nearshore and estuarine areas. This acceleration is mainly driven by the water depth and bottom morphology rather than by wind effects because similar patterns are observed at 14.30 and at 23.30. However, the 100-m spatial resolution of
the employed computational grid smooths the bathymetry and thus some key processes as the depth-induced refraction of waves (e.g. Komen et al., 1994) are likely misrepresented. This low resolution is not appropriate for studying the nearshore dynamics. Off the BoS, a higher spectral surface Stokes velocity is found with respect to its monochromatic value at the time when the bi-frequency spectrum is observed (14.30). By contrast, at 23.00 the spectral velocity is smaller than the monochromatic one, but the difference is negligible, in line with the observed mono-frequency spectrum. Generally these differences are rather
homogeneous across the entire computational domain.

The near-bed wave orbital velocity computed by WAVEWATCH-III coupled to CROCO v1.2 (UBRs from run 2WF) and its monochromatic approximation computed by CROCO v1.1 (UBRm from run 2WC) are compared for two time instants ( Fig. 19). These results indicate that an increase of spectral near-bottom wave orbital velocity with respect to the monochromatic values is present offshore and mainly depends on the nature of offshore incoming wave spectra and wind forcing. Conversely,
and differently from the Stokes drift, bathymetric effects in the surf zone leads to a decrease of spectral wave orbital velocity that reduces to the value predicted by the Airy theory or attains even a smaller value. Also in the case of the wave orbital velocity, the larger differences between spectral and monochromatic values are observed in the surf zone within the bay under both forcing conditions. This confirms that these differences are not only associated to the nature of the forcing spectra, but also on the wave propagation in shallow waters.





## 5 Conclusions


In this study we have described the implementation and assessment of an improved coupling (CROCO v1.2) between the oceanic model CROCO and the spectral wave model WAVEWATCH-III which includes newly exchanged spectral wave fields used to compute wave forcing terms in the wave-averaged governing equations. In addition to significant wave heights, mean wave periods and directions sent from the wave model to the circulation model in CROCO v1.1, we have added the sending

of mean wave length, near-bottom wave orbital velocity, surface Stokes drift, Bernoulli head, and wave-to-ocean energy flux. Then, these newly exchanged fields are used instead of their monochromatic approximations to compute wave-induced pressure gradients, non-conservative wave effects, and wave-enhanced vertical mixing, including the terms relevant for the surf zone. The impact of using wave forcing terms computed over the full spectrum on the model solutions has been assessed from the coastal dynamic point of view for a regional configuration of the Bay of Somme, which is dominated by macro-tidal currents

that are influenced by storm winds and waves.

Model results have been compared against those of an existing wave model and in-situ wave and current measurements. Modelled waves are on average of the right order of magnitude, but minimum height are overestimated and maximum heights are underestimated, and a phase shift is observed throughout the whole simulated time-series. The biases are however similar to those of the wave hindcast used to force the model boundaries, which also uses similar parameterizations. These biases

may thus be attributed to their propagation from the boundaries, and/or parameterization setup. Differently, the fair matching between modelled water levels and currents with both current measurements from punctual (AQDP) and profiler (ADCP) current-meters validates the new numerical developments in the circulation model. The modelled hydrodynamics well captures the macro-tidal range off the bay. Sensitivity experiments to the forcing indicate that metocean forcing does not significantly affect water levels and near-bottom currents that are primarily dominated by tides. Conversely, wind- and wave-induced effects

are found to regulate the currents at the free-surface. Here, storm winds blowing onshore accelerate flood flows and decelerate ebb flows, modifying the overall logarithmic vertical profile of tidal currents. Concurrently, storm waves act decreasing at the surface wind-driven acceleration, smoothing the vertical current profile throughout the entire water column. Modelled currents are correctly predicted during most of the phases of the tidal cycle, particularly at current reversal. Only at the peak of tidal floods a small phase shift between modelled and measured currents is observed. However, this slight periodic mismatch is

found to do not depend on random metocean forcing and thus it did not prevent the testing of our model developments.

The additional spectral wave-induced terms significantly affect the results in the case of storm waves and winds opposed to tidal flows, reducing the wave-induced deceleration of the vertical profile of tidal currents. Their contributions provide current magnitudes closer to measurements than those predicted using their monochromatic formulations, particularly at the free surface. In the particular case of the Bay of Somme, flood (ebb) surface currents are overestimated (underestimated) during

passing storms when approximating these spectral fields with their monochromatic counterparts. The error associated with this approximation is the largest when winds and tidal flows are opposed, and when the wave spectrum is not mono-modal. Among the investigated, additional spectral wave-induced terms the surface Stokes drift and the near-bed wave orbital velocity are the most impacting ones. The spectral Stokes drift leads to an increased advection towards the free surface, while the spectral



near-bed wave orbital velocity leads to a shifting of the vertical profile of currents close to the seabed. Their importance is
also thought to increase towards shallow waters where winds and waves dominate the nearshore circulation, with implications
on air-sea interactions and sediment transport processes. As such, our model development provides a better description of the
competing effects of tides, winds and waves on the oceanic circulation in coastal macro-tidal areas, and is now available in
CROCO v1.2 for future studies.

*Code and data availability.* The two exact versions of the CROCO model used to produce the results presented in this study are archived on
Zenodo (https://doi.org/10.5281/zenodo.7415343), as are simulation configurations, input files, in-situ data used for assessing the simulations
and codes to run the model and generate the figures of the paper (https://doi.org/10.5281/zenodo.8046629). The official repository of the
WAVEWATCH-III code can be found at https://github.com/NOAA-EMC/WW3/releases/tag/6.07.

*Author contributions.* ACB and FD planned the model developments and the modelling campaign; GP, MB, and ACB performed the setup,
validation, and applications of the developed model; SLB provided the ADCP and AQDP data, while GP, MB, and ACB analysed them; GP
and ACB wrote the manuscript draft; All authors reviewed and edited the submitted version of the manuscript.

*Competing interests.* The authors declare that they have no conflict of interest.

*Acknowledgements.* This study was funded by SHOM (French Navy Hydrographic and Oceanographic Service) in the framework of the
DEMLIT research project. Current data acquisition benefited from funding from Région Haute-Normandie and SHOM (reasearch contract
relative to dune dynamics in a under-saturated environment). The numerical modelling was performed using computational resources from
CRIANN (Normandy, France). Authors also thank the CROCO development team.



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



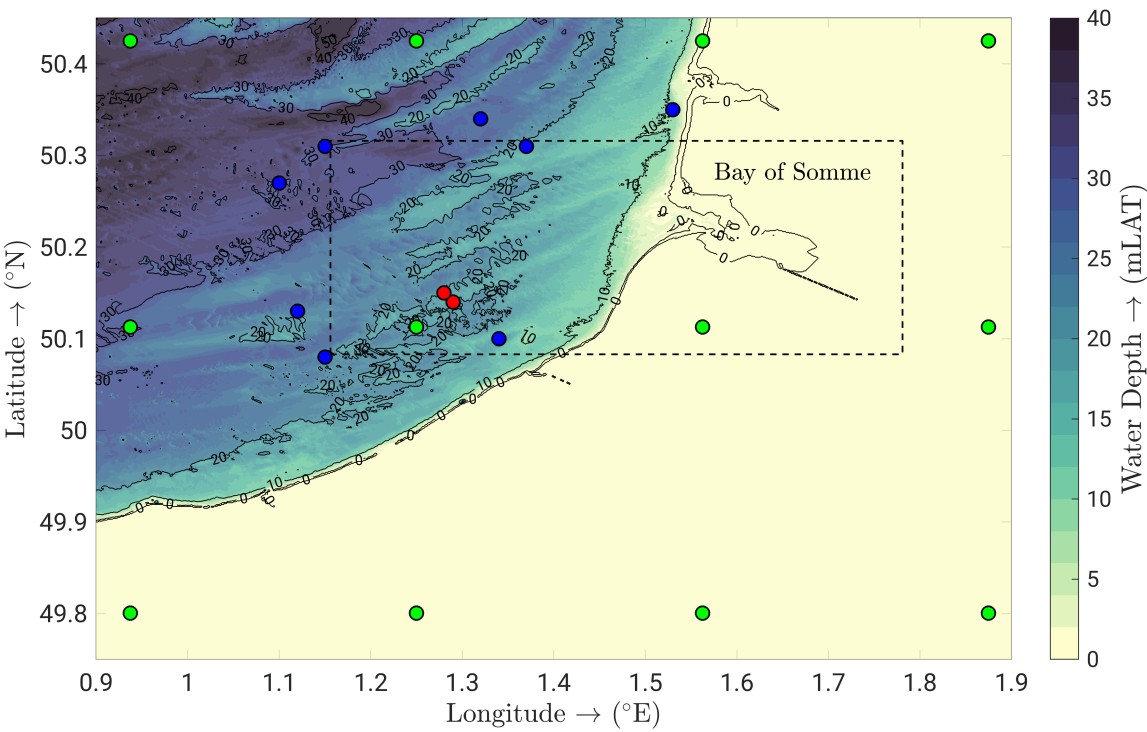

**Figure 1.** Bathymetric map of the study area. Dotted line encloses the computational domain. Green and blue marks indicate respectively the locations of CFSR global reanalysis wind data provided by NCEP and the HOMERE wave spectral data from which wind and wave forcing are interpolated over the computational grid and its open boundaries. Red marks indicate the monitoring stations (ADCP C3 to the north-west and AQDP B to the south-east).



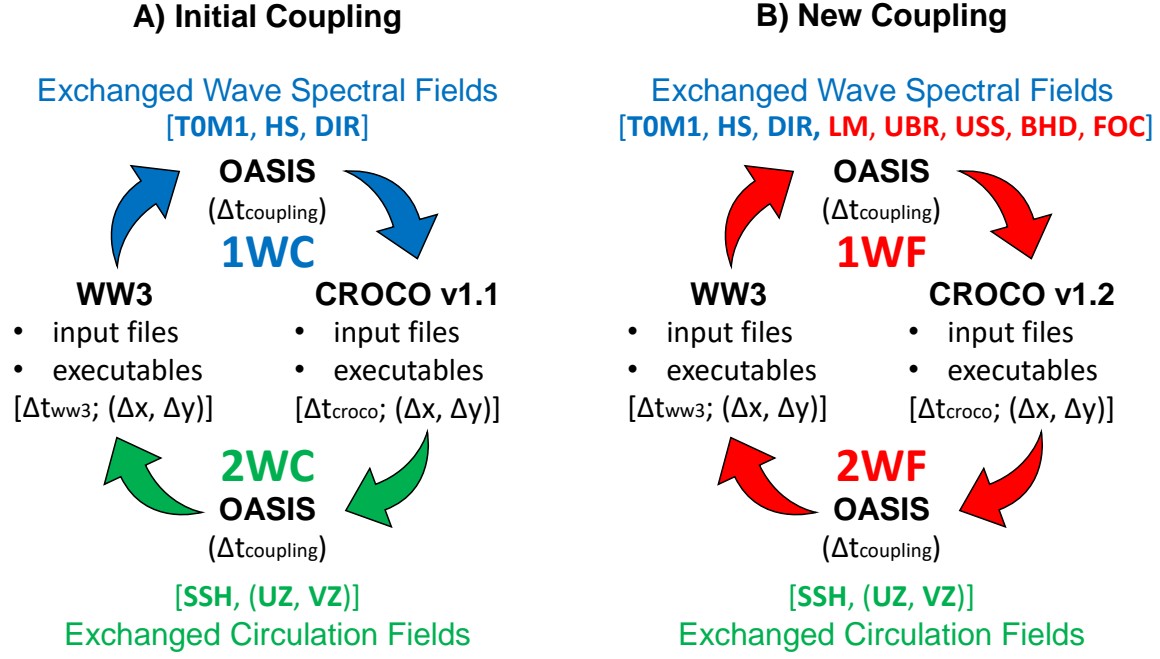

**Figure 2.** Schematic representation of the coupling between CROCO and WAVEWATCH-III using OASIS. Available coupling procedures in CROCO v1.1 are summarized on the left. Only three mean spectral wave fields (blue labels) are provided by WAVEWATCH-III in the one-way coupling (1WC), i.e., mean wave period ($T_{0M1}$), significant wave height ($H_S$), and mean wave direction ($\theta_w$), while in the two-way coupling (2WC) CROCO provides circulation fields (green labels), i.e., water levels (SSH) and surface currents (UZ,VZ). The newly developed coupling procedures in CROCO v1.2 are summarized on the right. In both one-way (1WF) and two-way (2WF), additional wave spectral fields (red labels) are exchanged, i.e., the mean wave length (LM), Bernouilli head pressure (BHD), wave to ocean energy flux (FOC), magnitude and direction of the surface Stokes drift (USS), and near-bottom wave orbital velocity (UBR).





**Table 1.** Summary of the fields exchanged within the OASIS coupler between CROCO and WAVEWATCH-III in each numerical simulation.

| Run ID | Description | $H_s$ | $T_{0m1}$ | $\theta_w$ | Ssh | $U_z$ | $V_z$ | Lm | Bhd | Foc | Uss | Ubr |
|--------|-------------|-------|-----------|------------|-----|-------|-------|-----|-----|-----|-----|-----|
| CRX | Only tidal forcing | | | | | | | | | | | |
| WND | Tidal and wind forcing | | | | | | | | | | | |
| 2WC | Two-way coupling (v1.1) | X | X | X | X | X | X | | | | | |
| 2WC+LM | Spectral mean wave length | X | X | X | X | X | X | X | | | | |
| 2WC+BHD | Spectral Bernoulli head | X | X | X | X | X | X | | X | | | |
| 2WC+FOC | Spectral energy flux | X | X | X | X | X | X | | | X | | |
| 2WC+USS | Spectral Stokes drift | X | X | X | X | X | X | | | | X | |
| 2WC+UBR | Spectral wave orbital velocity | X | X | X | X | X | X | | | | | X |
| 2WF | Two-way new coupling (v1.2) | X | X | X | X | X | X | X | X | X | X | X |

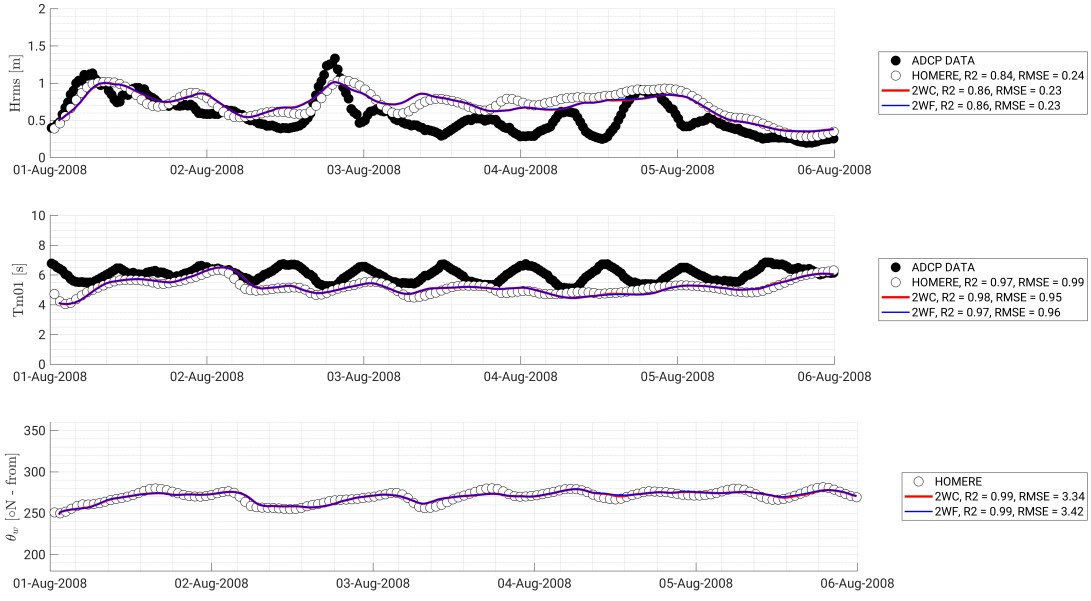

**Figure 3.** Time-series of computed mean wave parameters. Comparisons of model results (solid lines) with in-situ measurements (ADCP data, black dots) and output of large-scale spectral wave model (HOMERE data, white dots) in terms of significant wave height (top), mean wave period (middle) and mean wave direction (bottom). Note that statistical metrics in the bottom-left insert correspond to the correlation between modelled directions and those predicted by the HOMERE hindcast (no information was available from the ADCP)





**Figure 4.** Root mean square wave height (coloured shading) with superimposed wavenumber vectors (black arrows) computed in simulation 2WF and bathymetric contours (white contours) at 2:30 p.m. on 2 August 2008. Top panel shows the entire computational domain, while bottom panel shows a focus (grey shaded large rectangle in top panel) in the vicinity of the ADCP and AQDP locations (black circles), where maximum and minimum water depths are 15 and 25 m, respectively. The grey shaded small rectangle in top panel indicates the area in which cross-shore wave height and set-up are extracted and plotted in Fig. 5.

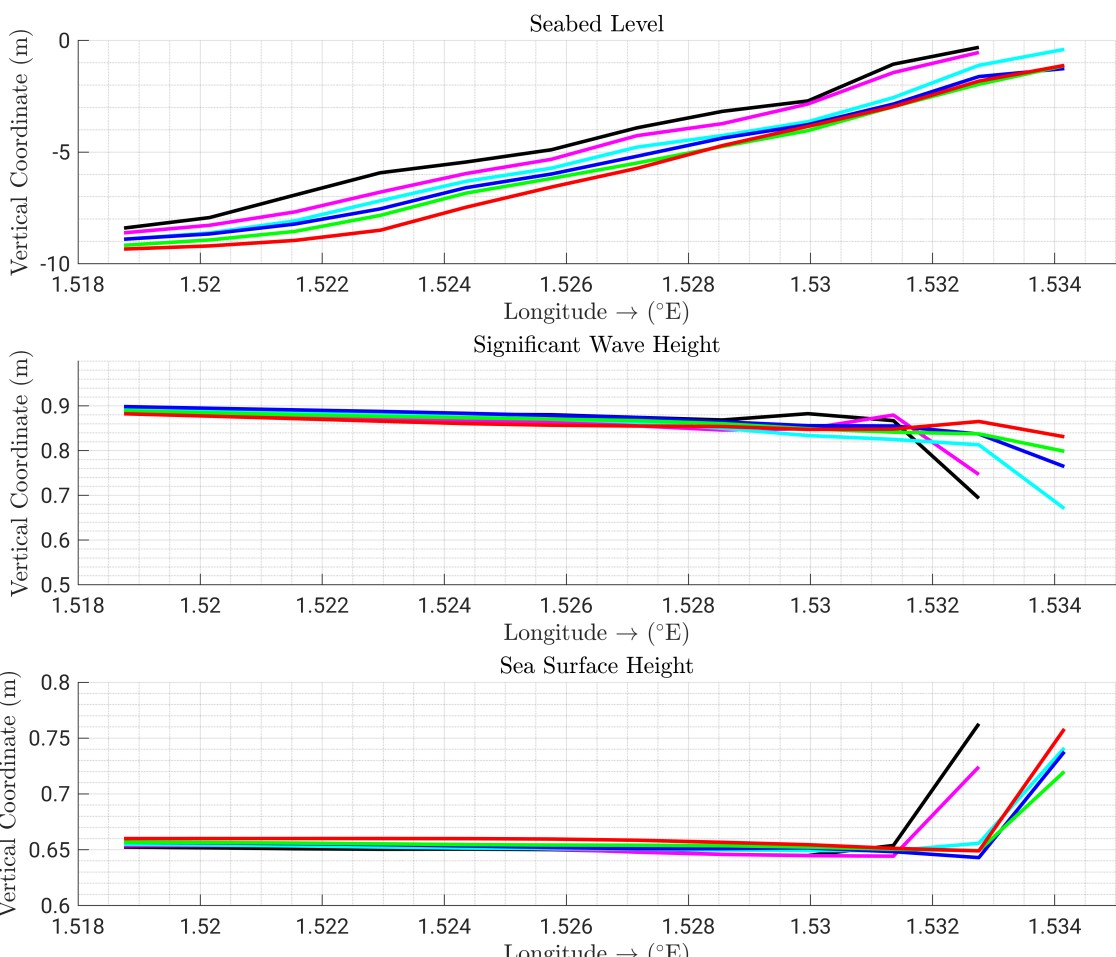

**Figure 5.** Cross-shore profiles computed from simulation 2WF at 2:30 p.m. on 2 August 2008 of the mean water depth (top), the significant wave height (middle), and water level (bottom) for 6 different transects respectively at latitudes 50.287°N, 50.289°N, 50.291°N, 50.293°N, 50.295°N, 50.297°N, varying in longitude between 1.519°E and 1.536°E ( Fig. 4).



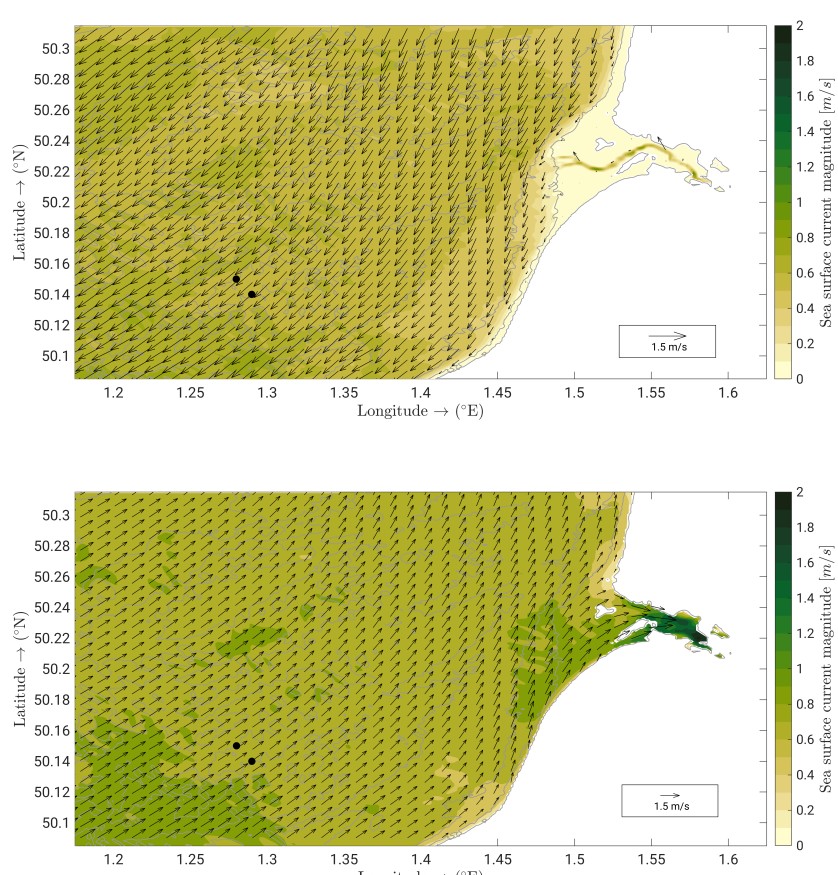

**Figure 6.** Surface current magnitudes (coloured shading) with superimposed current vectors (black arrows) computed from simulation 2WF and bathymetric contours (grey contours) at the peak of the ebb tidal flow the 2 August 2008 at 5 pm (top) and that of the flood six hours later (bottom). Black circles are the ADCP and AQDP locations.

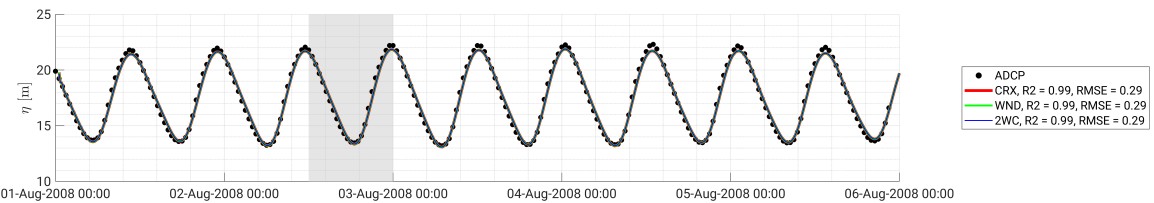

**Figure 7.** Comparison of modelled water levels (solid lines) with in-situ measurements (dots) at the ADCP location. Shaded area shows period of fresh to strong winds with moderate to large waves in the investigated time window.




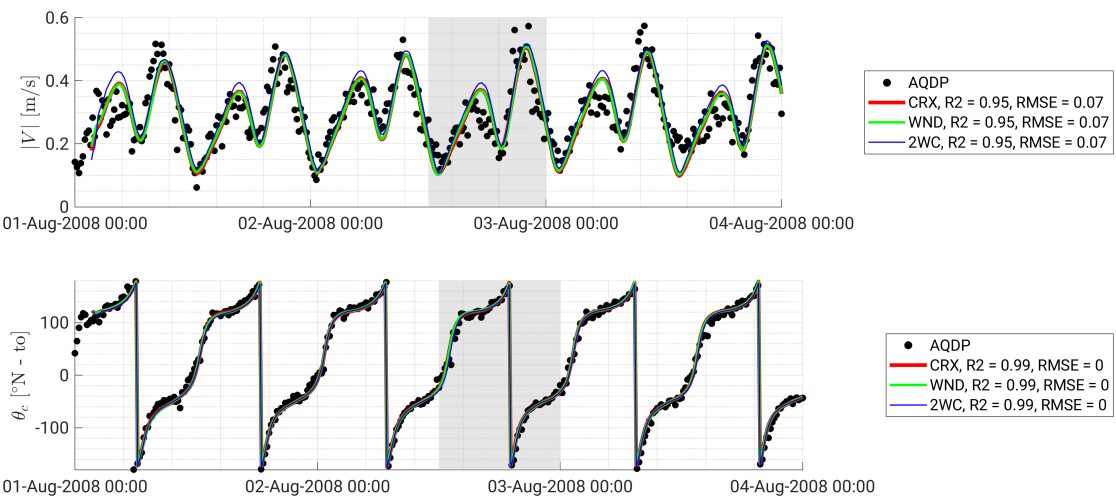

**Figure 8.** Time series of computed near-bottom tidal currents. Model results (lines) are compared with AQDP (dots) measurements in terms of (top) current magnitude and (bottom) 'going to' direction at 1 m above bottom (mab). Shaded areas show the period of fresh to strong winds with moderate to large waves in the investigated time window.

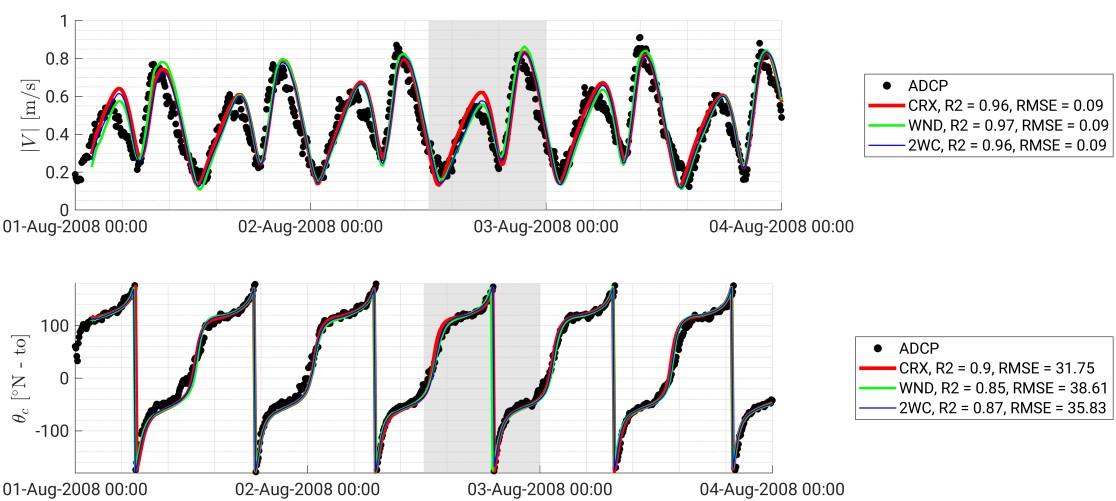

**Figure 9.** Same legend as for Fig. 8 but it refers to the tidal currents at 1 m below the free-surface (mbfs).



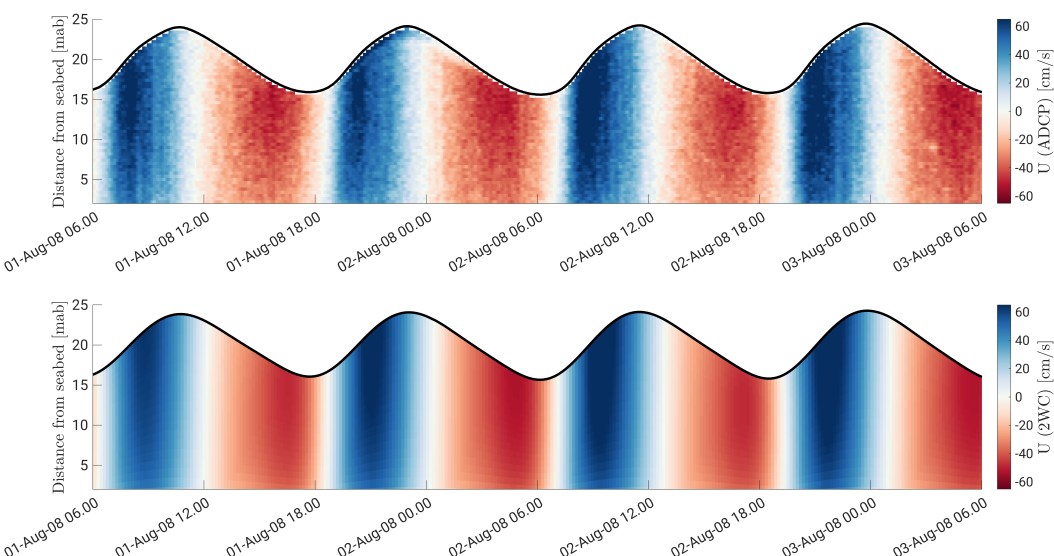

**Figure 10.** Time series of the measured (top) and modelled (bottom, 2WC) eastward tidal current velocity over the depth and at the ADCP location. Thick black line represents the free-surface elevation from the seabed in time.

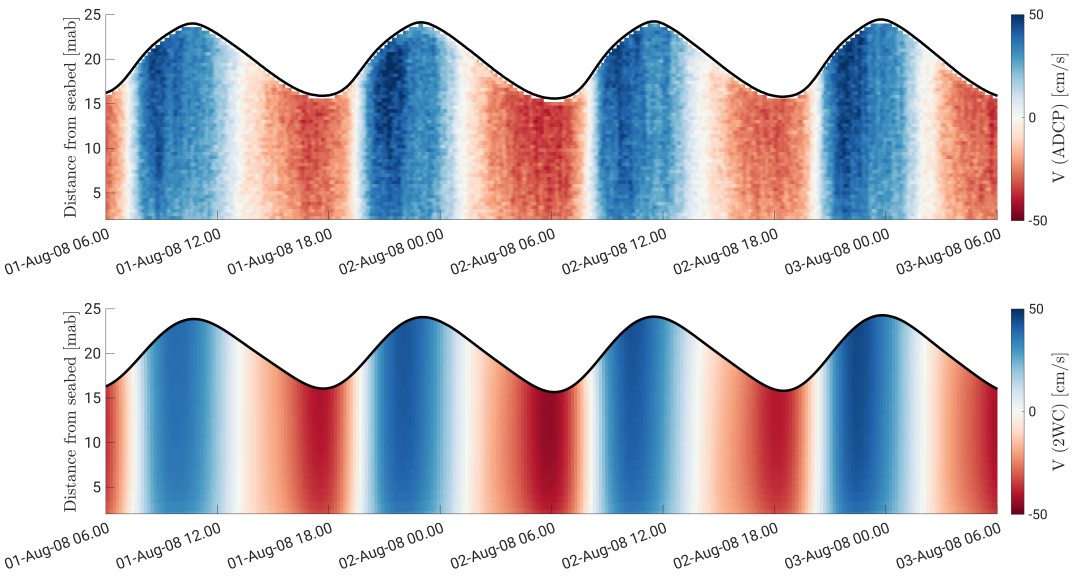

**Figure 11.** Same legend as for Fig. 10 but for the northward tidal current velocity.





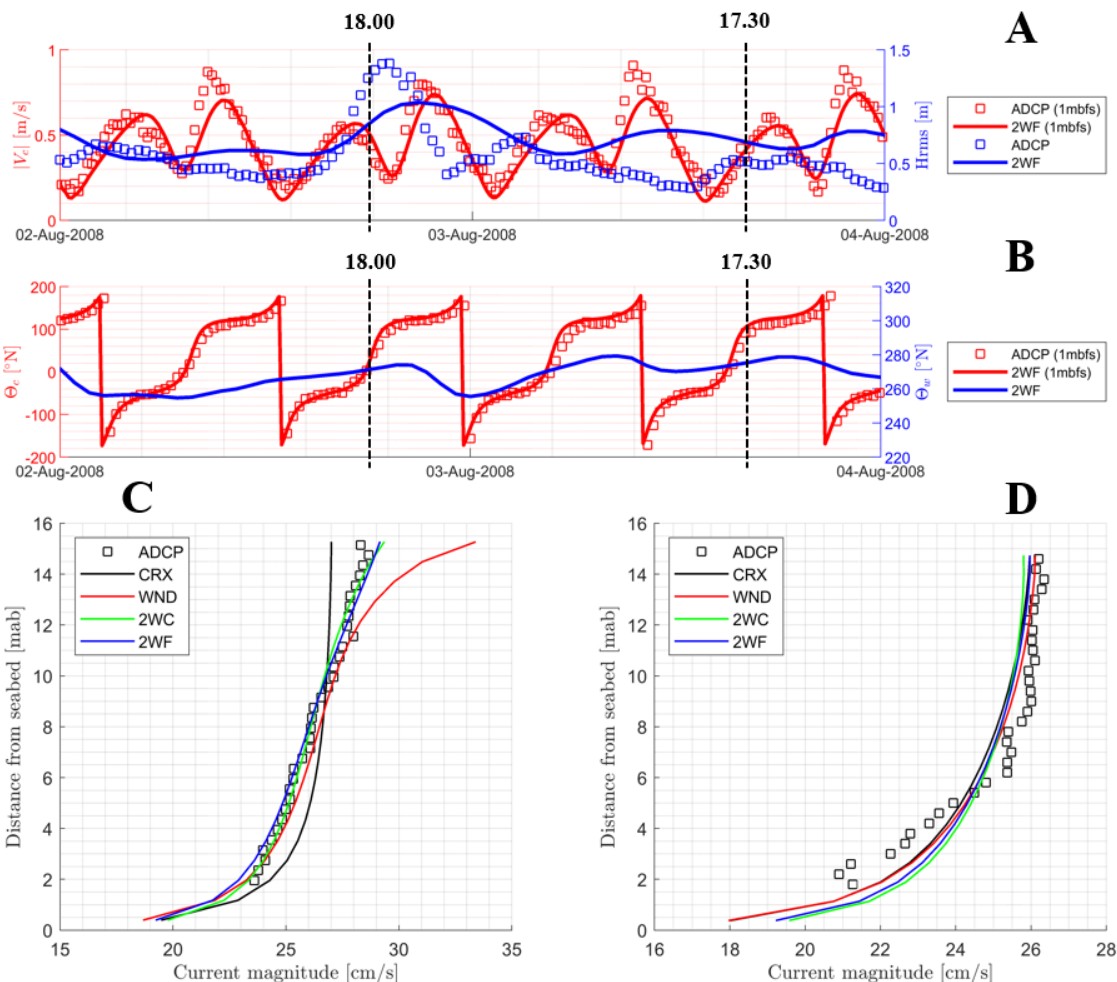

**Figure 12.** (A): time-series of measured (squares) and modelled (continuous lines) surface current magnitude (red) and wave heights (blue) at the ADCP location. (B): time-series of measured (squares) and modelled (continuous lines) surface current (red) and wave (blue) directions at the ADCP location. Dotted black lines indicate the two time instants at which vertical velocity profiles (bottom panels) are extracted, respectively for $H_{rms}$ larger (left) and lower (right) than 1 m. (C): measured (squares) and modelled (continuous lines) vertical profiles of the current on 2 August 2008 at 6 pm. (D): same legend as for (C) but refers to 3 August 2008 at 5.30 pm.



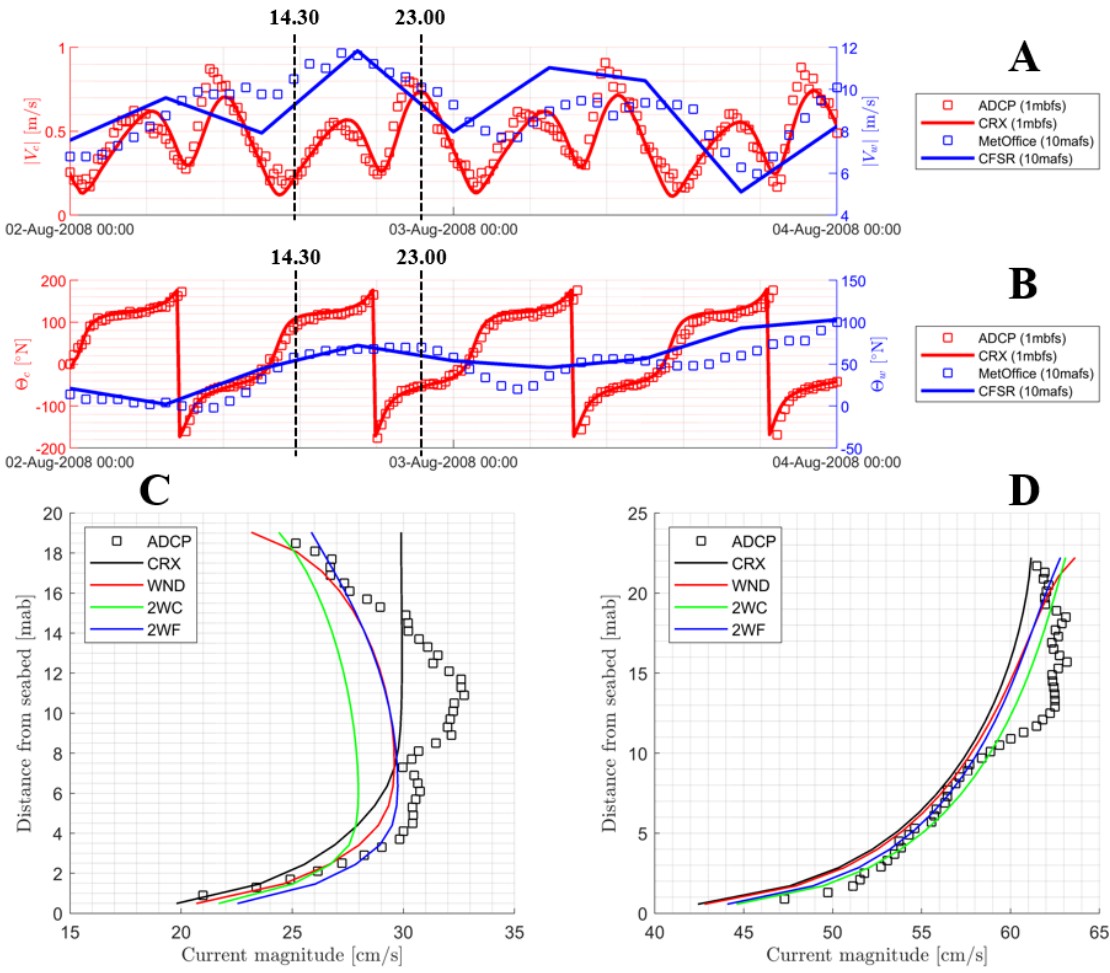

**Figure 13.** (A): time-series of measured (squares) and modelled (continuous lines) surface current magnitude (red) and wind speeds (blue). (B): time-series of measured (squares) and modelled (continuous lines) surface current (red) and wind (blue) directions. Dotted black lines indicate the two time instants at which vertical velocity profiles (bottom panels) are extracted which correspond to winds coming from southwest with speeds larger than 10 m/s, respectively during tidal ebb (left) and flood (right). (C): measured (squares) and modelled (continuous lines) vertical profiles of the current on 2 August 2008 at 2.30 pm (tide reversal). (D): same legend as for (C) but refers to 2 August 2008 at 11 pm (tidal ebb).



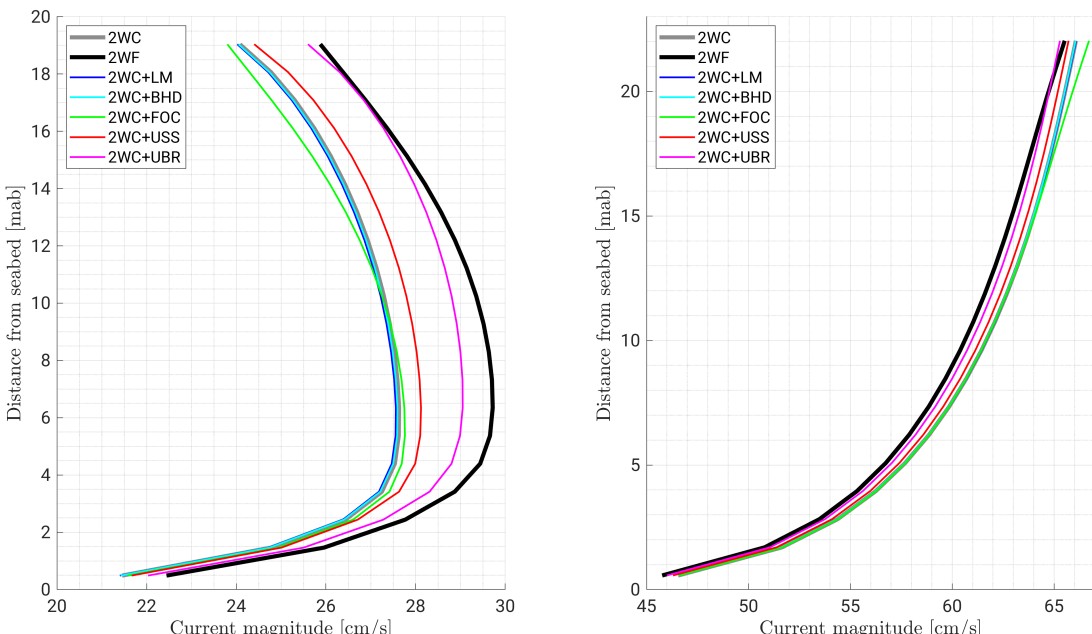

**Figure 14.** Contributions of the different newly exchanged spectral fields to the vertical current profile in the case of winds coming from south-west with speeds of the order of 10 m/s, respectively during tidal ebb (left) and flood (right). Modelled profiles are extracted at the ADCP location the 2 August 2008 at 2.30 pm (left) and 11 pm (right), corresponding to the profiles shown in Fig. 13.





**Figure 15.** Wave energy spectra computed by the model at the ADCP location the 2 August 2008 at 2.30 pm (A,C) and 11 pm (B,D), corresponding to the profiles shown in Fig. 13 and 14. Top panels (A,B) show frequency-direction spectra, while bottom panels (C,D) show frequency spectra integrated over directions.





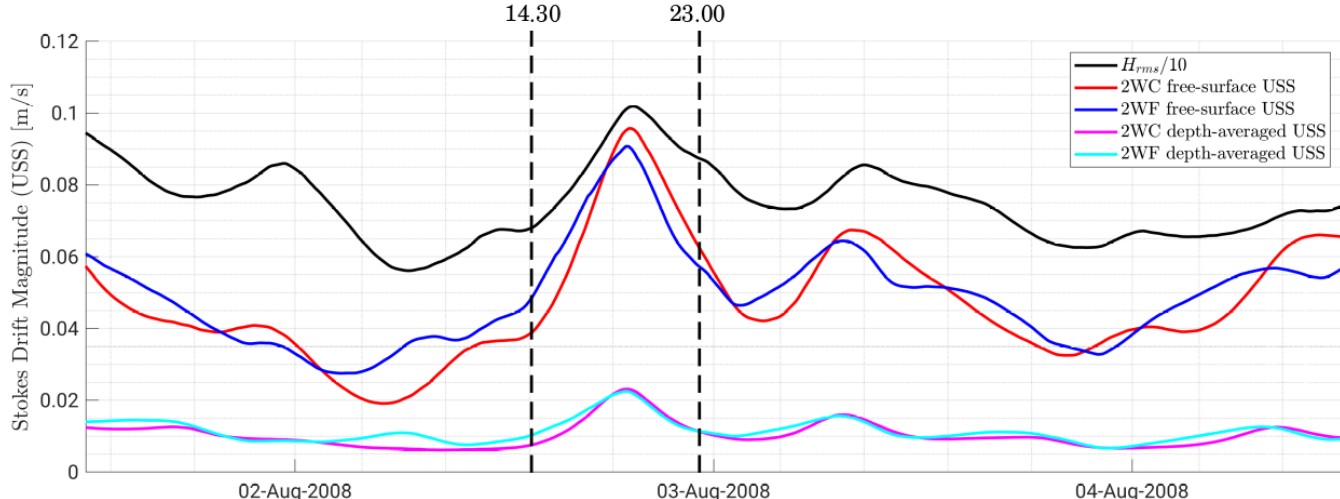

**Figure 16.** Time-series of the modelled magnitude of the Stokes drift at the ADCP location. Black line represents the $H_{rms}$. Red and blue lines are the modulus of the free-surface Stokes drift respectively computed by CROCO v1.1 and v1.2. Magenta and cyan lines are the depth-averaged values respectively of the monochromatic and spectral Stokes drift.

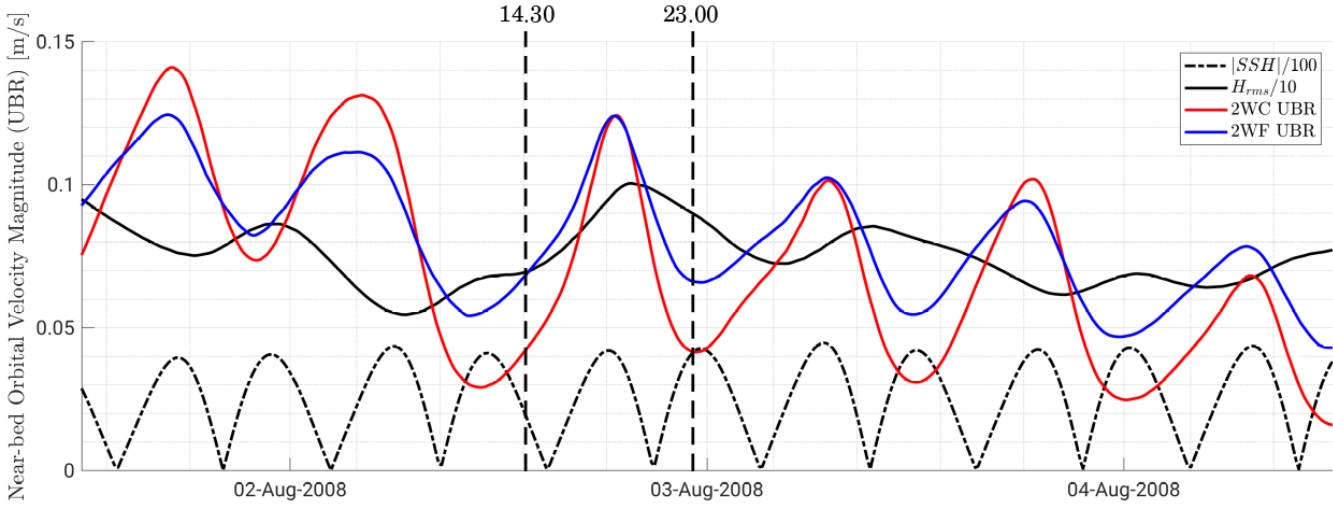

**Figure 17.** Time-series of the modelled magnitude of the near-bed wave orbital velocity at the ADCP location. Black line represents the $H_{rms}$, while dotted line represents the absolute value of the sea surface height. Red and blue lines are the modulus of the near-bed wave orbital velocity respectively computed by CROCO v1.1 and v1.2.



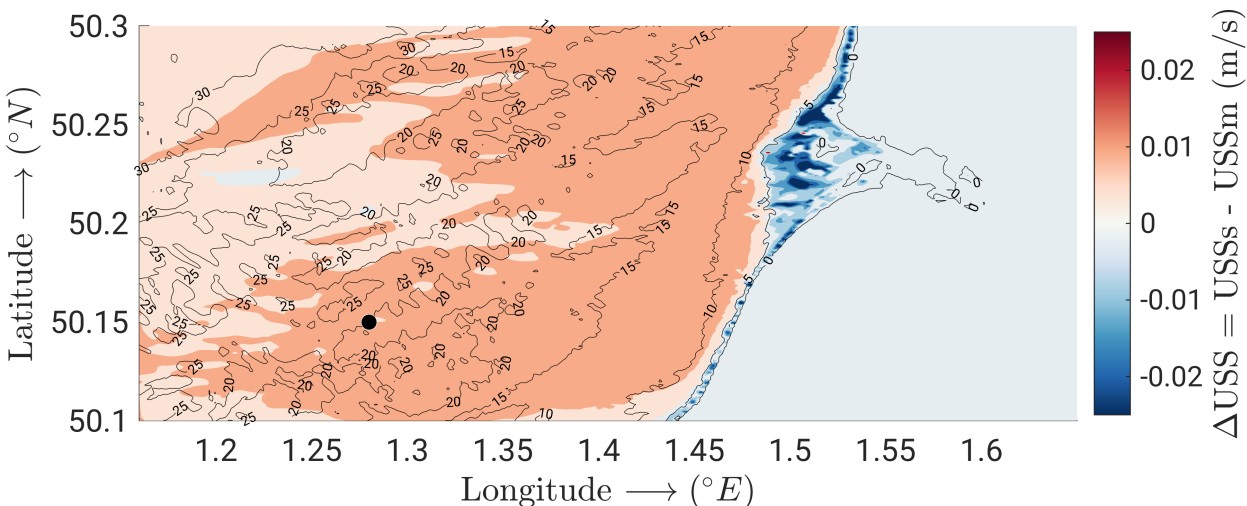

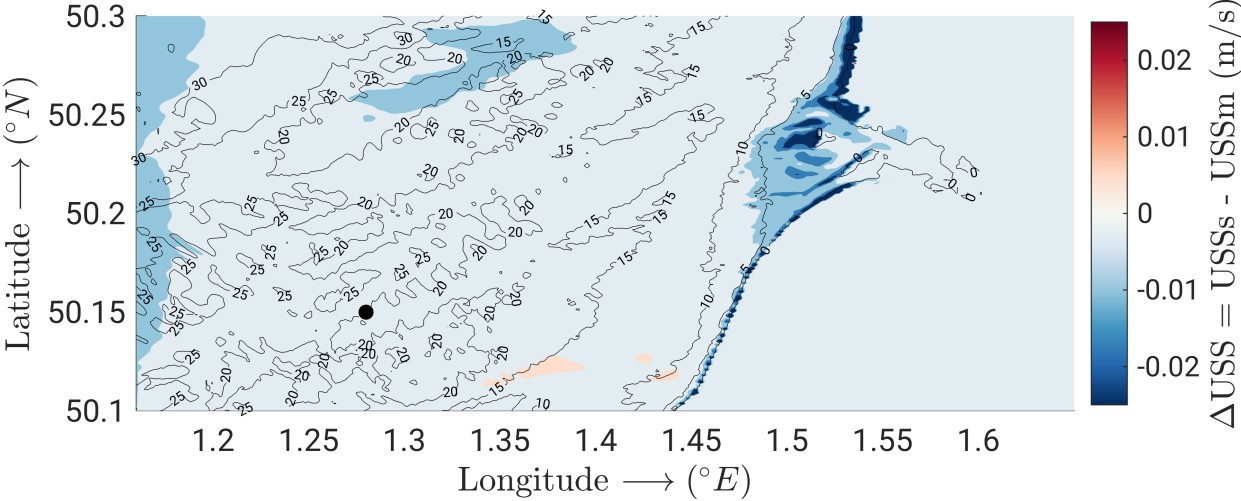

**Figure 18.** Snapshots of the difference between the spectral (USSs, 2WF) and monochromatic (USSm, 2WC) Stokes drifts respectively computed by CROCO v1.2 and v1.1 across the whole model domain. Top panel shows results at 2.30 pm of the 2 August 2008, while bottom panel those at 11 pm of the 2 August 2008. Red marker indicates the ADCP location.





**Figure 19.** Snapshots of the difference between the spectral (UBRs, 2WF) and monochromatic (UBRm, 2WC) near-bed wave orbital velocity respectively computed by CROCO v1.2 and v1.1 across the whole model domain. Top panel shows results at 2.30 pm of the 2 August 2008, while bottom panel those at 11 pm of the 2 August 2008. Red marker indicates the ADCP location.