# Peer review of "Implementation of additional spectral wave field exchanges in a 3D wave-current coupled WAVEWATCH-III (version 6.07) - CROCO (version 1.2) configuration and assessment of their implications for macro-tidal coastal hydrodynamics"

_EGUsphere, 2023_

## Author Comment (AC1)

**Reply to Reviewer 1**

**General comment**

*This paper presents a new coupling procedure between the 3D ocean circulation model CROCO and the spectral wave model WAVEWATCH-III, handled through the OASIS-MCT coupler, in order to improve the representation of wave-current interactions in the coastal region. More precisely, efforts are made to incorporate wave quantities derived from the full (bi-dimensional) wave spectrum that better represents "real" directionally and frequency-broad wave fields. This is in contrast with "monochromatic" approximations that use a representative wavenumber (e.g. mean or peak) as originally used and implemented in ROMS (Uchiyama et al., 2009, 2010).*

*After a broad overview of the different approaches for simulating wave-current interaction in the coastal context, the paper describes the new forcing terms. The rest of the paper focuses on a series of numerical experiments performed in the geographical setting of the Bay of Somme, France, a macrotidal site located in the English Channel. In situ data (vertical current profiles at two locations) are used to assess the model performances as well as to evaluate the added value of the newly-proposed coupling procedure.*

*I found the manuscript relatively well organised, though not particularly well written. At least, it could have used a few more readings by the authors: e.g. some internal notes made by the authors remain in the core of the manuscript (see at lines 102-107), which is not acceptable in my opinion. The efforts to incorporate full spectral representation of wave quantities are welcome, as it should help to get a better and more realistic representation of wave forces while modelling wave-current interactions with this modelling system. The (scientific) novelty is not obvious, though, since similar efforts made by other authors in ROMS are overlooked, and such spectral representation already exists in other modelling systems. So it is not clear to me whether this contribution justifies a publication or not. I have several major comments on the present work that, I think, should be addressed before a resubmission.*

We express our sincere appreciation to the Reviewer for conducting a thorough review of our manuscript and for providing valuable feedback that has greatly contributed to the enhancement of our work. We fully understand the importance of addressing the major comments raised by the Reviewer before resubmission.

In terms of the organization and clarity of the manuscript, we acknowledge the Reviewer's valid point about the need for further editing and refinement of the text. We would like to clarify that the issue raised in lines 102-107 occurred during a minor revision phase, during which we were specifically instructed to change only the title and availability statement. Regrettably, an erroneous copy-paste during the final submission inadvertently included internal notes from another manuscript. We deeply regret this oversight. However, we want to assure the Reviewer that, apart from this isolated issue, the manuscript underwent multiple reviews and revisions by the authors prior to the initial submission to ensure its quality and coherence.

We concur with the Reviewer that certain relevant prior work was initially overlooked. Building upon the Reviewer's citations, we will incorporate references to pertinent previous studies and emphasize the specific contributions of our study in light of these works, particularly in terms of our utilization of full spectral representation of wave quantities. We also recognize the importance of highlighting the unique aspects of our work. To achieve this, we will revise the introduction to better underscore the novelty and distinctiveness of our study within the broader context of wave-current interaction modeling. This will encompass a discussion of how our approach differs from previous efforts, both within our modeling system and in other modeling systems. We firmly believe that our study provides valuable insights and advancements in the field of wave-current interaction modeling, and we are committed to conveying this effectively in the revised manuscript.

We are fully committed to address these comments and resubmit a revised manuscript that meets the necessary standards for publication.

**Major comments**

1) *The manuscript completely ignores many relevant and recent works that have looked into these aspects, some within the frame of the same modelling system. Within ROMS, from which CROCO is derived, the works by L. Romero and the UCLA team are completely overlooked (in particular Romero et al., 2021; Hypolite et al., 2022). This also includes the work initiated by N. Kumar (Kumar et al., 2017; Liu et al., 2021) that has evaluated the impact of the definition used for the Stokes drift velocities on the modelling of wave-current interactions at regional scales. The present manuscript also ignores modelling systems already incorporating full spectral wave quantities such as MOHID (Delpey et al., 2014) or SCHISM, which has been used to simulate and analyse the wave-induced nearshore circulation in numerous realistic settings at both regional (Guérin et al., 2017; Lavaud et al., 2020; Pezerat et al., 2022) and local (Martins et al., 2022) scales.*

We highly appreciate the valuable comment provided by the Reviewer, which has guided us in improving the quality of our manuscript with the refinement of the referenced literature and the introduction of our model developments. In response to the Reviewer's observations, we have addressed the omission of references to previous model developments within the same modelling framework and those from other model systems. Of particular significance is the inclusion of reference to the work initiated by Kumar et al. (2017). These references have been thoughtfully incorporated into the introduction of our manuscript to provide a more comprehensive context for our research.

To effectively underscore the unique contributions of our manuscript in relation to the works mentioned by the reviewer, we provide a detailed discussion of each specific work, highlighting both connections and distinctions with our study.

**Works Within the Same Modelling System**

*Kumar et al. (2017)* proposed a significant advancement in modelling wave-current interactions through the advocacy of spectral reconstruction for the forcing and coupling of wave-current models. This approach, rooted in the partitioning of WaveWatchIII, delivers more accurate estimates of the Stokes drift when compared to the spectral peak monochromatic approximation of *Uchiyama et al. (2010)*, which often results in underestimations. Later, *Liu et al. (2021)* confirmed the need of spectral estimates of the Stokes drift for both deep- and intermediate-water applications. Our study aligns with this methodology by employing full spectra from a regional spectral model to force our wave model and incorporating spectral estimates of the Stokes drift. Moreover, we have introduced additional spectral field exchanges to further refine the description of wave-current interactions, thereby distinguishing our novel approach.

*Romero et al. (2021)* introduced a novel ROMS WEC (Wave Effects on Currents) framework that extended the monochromatic approach of *Uchiyama et al. (2010)*. This extension encompassed spectral approximations of Stokes drift, Bernoulli head, and wave-induced vertical mixing, achieved through iterative computations and a switching mechanism between deep- and shallow-water formulations. *Hypolite et al. (2021)* used this framework to explore the impacts of the spectral approach on ocean mesoscale circulation variability, revealing relatively modest wave effects despite larger Stokes drift estimates. In consonance with their work, we have incorporated the exchange of spectral Bernoulli head in our model coupling. Additionally, our enhancements encompass novel spectral field exchanges designed to provide a more precise representation of the wave-induced bottom boundary layer. This includes the exchange of spectral bottom wave orbital velocity, which we have demonstrated to have a comparable impact on macro-tidal current profiles to that of the Stokes drift in our specific application. Notably, our study accentuates the significant influence of wave effects on macro-tidal currents, further distinguishing our research from prior investigations.

**Works on Different Model Systems Incorporating Full Spectral Wave Quantities**

While the references cited by the Reviewer describe developments akin to ours, all aimed at achieving a more comprehensive description of wave-current interactions, our study delves into aspects specific to the employed modelling system, setting it apart from the frameworks mentioned by the Reviewer in several key ways.

Regarding technical aspects, our coupled model uses different hydrodynamic and wave spectral models than the cited systems. These models are not integrated with each other, requiring a third model for coupling and thus introducing associated complexities and advantages. To do so, the CROCO coupled system uses the OASIS-MCT coupler, which is a set of libraries allowing for parallel exchanges and grid interpolations between different models. This framework has multiple advantages such as the possibility of choosing different grids for the different models (and eventually different nesting strategies), the possibility of choosing the coupling frequency, the exchanged variables, etc. This interface is thus particularly flexible and allows coupling CROCO with any model included in the OASIS-MCT library. Furthermore, the computational grids used by CROCO (which is not necessarily the same of the wave model) are structured rather than unstructured as the cited modelling systems.

Regarding historic aspects, ROMS was originally designed for regional oceanic applications. CROCO inherited from ROMS and recent model developments added up new capacities to link the regional, coastal and nearshore scales. Our work, in line with these recent advances, concentrates on its adaptation for intermediate-water macro-tidal conditions, distinguishing it from the cited modelling frameworks which are more nearshore-oriented.

These distinctions underscore the specific contributions and relevance of our study within the context of the CROCO modelling framework and the broader field of wave-current interactions.

2) *The present numerical setup is far from being ideal to discuss the added value of (full) spectral representation of wave quantities, for several reasons. The hydrodynamics simulated here is overly controlled by the forcing used at boundaries, both in terms of water levels, currents and waves, to the point that the reader asks himself what is the added value of using a local model. For instance, the wave results at the measurement site show no improvements over the large scale model used as forcing. Furthermore, water levels and currents are (to my understanding) forced from a 2D hydrodynamic model. Considering the importance given to the vertical shear, how can we be sure that the boundary is taken sufficiently far away from the location where measurements were obtained and vertical velocity profiles discussed? I suspect that the boundary is too close to the measurement stations in order to discuss differences of the order of 1 cm/s at the top layer of the water column.*

We sincerely appreciate the reviewer's valuable comments, which have raised important concerns regarding the numerical setup and the discussion of the numerical results to assess the added value of employing a full spectral representation of wave quantities.

In our research, we used a numerical model that incorporates wind stresses at the free surface, tidal currents and levels as well as full wave spectra at the offshore boundaries to simulate the hydrodynamics within the macro-tidal bay of Somme in the English Channel. This choice of setup was grounded in our belief that these specific forcing terms are the most pertinent factors for capturing the dynamics of the study area.

We duly acknowledge the reviewer's concern regarding the influence of wave boundary conditions on our simulation. It is indeed true that the hindcast data used to force the wave model exhibit some biases in our study area. Nevertheless, it is crucial to emphasize that the use of full spectra at the boundaries was made because is the best (and most computationally demanding) practice in modelling spectral waves, particularly when investigating complex phenomena such as spectral Stokes drift and wave-current interactions (as highlighted in Kumar et al., 2017, and Liu et al., 2021, both now cited in the revised manuscript).

Addressing the concern related to water levels and currents, we acknowledge the possibility that the proximity of the southern boundary to the measurement sites could result in an appearance of excessive control by the imposed forcing. To investigate this, we conducted an additional numerical simulation with an extended computational grid in the south and west directions, effectively placing the measurement sites further from the open boundaries. In this

new simulation, we limited the forcing to only the west and north offshore boundaries to assess the independence of the hydrodynamic model results from the relative distance between the forcing boundary and the measurement locations (refer to Figure 1).

The results of this supplementary simulation align closely with the outputs of the original simulation which solely employed tidal forcing. This congruence is evident in terms of circulation patterns (as illustrated in the tidal flood snapshot, Figure 2, which can be compared with Figure 6 of the submitted manuscript), time series of near-bottom currents (Figure 3), and discussed vertical profiles (Figure 4).

These additional results convincingly demonstrate that the boundary was situated at a sufficient distance from the measurement locations where vertical velocity profiles were examined. It is vital to emphasize also that the discrepancies observed in the modelled tidal currents between different tidal forcing setups are approximately one order of magnitude lower than those found when comparing pure tidal profiles with those influenced by wind and waves (as evident if comparing Figure 4 with Figure 13 of the submitted manuscript), particularly as we approach the free surface.

Based on these additional results, we assert with confidence that the choice of tidal forcing in our simulations does not exert undue control over the modelled hydrodynamics. This support our validation of the macro-tidal currents influenced by wind and waves in the bay of Somme and the discussion of the substantial added value derived from the incorporation of additional spectral wave field exchanges.

3) *The discussion of wave processes, particularly in shallow water depths, is in my opinion weak throughout the paper. Discussion on depth-induced refraction (short waves in relatively deep water depth) is not supported by the numerical results, while the section dealing with wave breaking in shallow water (set-down and setup) is really weak, to say the least. Reading that wave effects are "removed" in depths less than 2 m (why is that so?), how can we then expect a wave setup to form near the shoreline? The influence of using a spectral representation is expected to be much more important in wave-dominated shallow-water environments, so we expect this study to also consider such environments where this model has already been applied (e.g. Duck N.C. in Uchiyama et al., 2010; or the Biscarosse site in Marchesiello et al., 2015).*

We concur with the Reviewer's perspective that the use of a spectral representation is more significant in wave-dominated shallow waters. In such environments, the effects of waves on currents can manifest with particular prominence, a factor that is less pronounced in intermediate water depths, as the locations of the measurements discussed in this study.

To provide a comprehensive context for our study and elucidate the rationale for our choice of modelling application, it is essential to place our research within the framework of its funding project. The DEMLIT project, funded by SHOM in collaboration with the University of Caen Normandy, is dedicated to investigate the sand filling dynamics in the Bay of Somme. The submitted manuscript represents the initial phase of this project, focusing on the macro-tidal

circulation off the bay and its interactions with offshore winds and waves. To do this we employed available tidal atlases and low-resolution bathymetric data from SHOM complemented by in-situ measurements from the University. It is important to note that we did not account for wave effects on currents below 2 meters water depth, as the interpolation of the low-res bathy data within the bay resulted in steep seabed gradients, leading to numerical instabilities in the surf zone.

This model will serve as a nest model, providing a foundation for higher-resolution, shallower-water nested applications. These forthcoming applications will explore hydrodynamics and sediment transport processes within the bay. Notably, we have collected new shallow-water, high-resolution bathymetric data and ADCP measurements in the surf zone to inform and validate this forthcoming phase of the modeling.

We understand the significance of accurately reproducing wave processes in shallow waters, especially while striving to improve the representation of wave effects on currents, and we acknowledge that previous versions of our model have been applied in such settings, as highlighted by Uchiyama et al. (2010) and Marchesiello et al. (2015). However, it is crucial to underscore the primary objective in our current study, which is centered on model the interactions of macro-tidal currents with offshore winds and waves. Given this clarification, we will remove from the section dealing with the assessment of modelled wave the last paragraph discussing wave breaking in shallow waters. These aspects, though undeniably important, will be reserved for the future extensions of our modelling efforts, as we focus on the fundamental objective of our study.

4) *Instead of a lengthy description of the field site, and given the chosen journal, we expect much more description on the coupling procedure: where there files modified in the wave or hydrodynamic models or can new versions of these models be pulled directly? Can the new coupling procedure be reproduced elsewhere easily? What about interpolation of the different fields and where is it handled?*

We appreciate the Reviewer's feedback and agree that a more detailed description of the coupling procedure is warranted.

All the additional spectral fields required for the proposed advanced coupling were already incorporated into the WAVEWATCH-III model, and we simply added the necessary exchanges in CROCO to compute the wave forcing terms. These modifications have been available in CROCO since the v1.2 release. Both the 'monochromatic' and 'full' coupling procedures are accessible for comparison in different configurations, and users can easily switch between them by activating or deactivating a preprocessing key.

The OASIS coupler handles all data exchanges, including interpolations on different grids. Coupling information is specified in a parameter file exclusively used by OASIS. Therefore, applying this new coupling to another configuration and assessing the impact of our developments should not pose significant challenges.

**More specific comments:**

ABSTRACT

1) *Lines 7-8: Given the presented results, and the comments above, I think that the "significant" currently needs to be toned down. Considering nearshore situation with adequate numerical experiments could surely better highlight the impact of using (full) spectral wave quantities.*

Following the Reviewer's suggestion, we will carefully reconsider the wording in the abstract to accurately reflect the main outcomes of our study while taking into account its limitations in terms of nearshore modelling and the primary scope of our work.

2) *Lines 15-16: What about nearshore environments, where Bernoulli heads and breaking acceleration terms will be the dominant forcing of the nearshore circulation?*

In response to the Reviewer's valuable feedback, we will carefully revise the text throughout the entire manuscript to eliminate any ambiguity regarding the study's objective. We will explicitly narrow down the study objective from 'modelling wave-current interactions in coastal areas' to 'modelling interactions of macro-tidal currents with winds and waves'. This adjustment will ensure greater clarity and alignment with the primary goals of our research.

INTRODUCTION

3) *Line 21: "Wave-induced currents and setup" are not forcing terms.*

The text will been corrected.

4) *Line 26: Reformulate with "wave-dominated environments"?*

As we clarified in our response to the third major comment, the specific focus of our study does not revolve around 'wave-dominated environments.' We will take extensive measures to overhaul the entire introduction section, ensuring that the text provides a clear and accurate representation of the paper's objective. This comprehensive revision will guide the reader toward a precise understanding of our research goals.

5) *Lines 34-35: "Three-dimensional modelling of the wave-induced flow was requested". I think that the authors can find much better context and needs for resolving 3D currents in the nearshore context (e.g. the transport of sediment, particles or tracers).*

The sentence will be modified in accordance with the primary objective of our study. We appreciate the Reviewer's suggestion, and it's essential to note that even in intermediate waters within macro-tidal environments, the resolution of the vertical current profile is crucial for understand and model the transport of sediment, particles, and tracers. While the effects of winds and waves may be comparatively less pronounced, the accuracy of the vertical current profile remains instrumental in capturing these important aspects even in the discussed intermediate-water setting.

6) *Line 35: Specify what these theories are used for.*

The text will be update to specify that these theories derive depth-dependent expressions of the wave-averaged flux of momentum due to waves to study the interaction of currents and waves in water of finite depth.

7) *Line 40: "With simplifications". The end of this section feels too light, and considering the scope of the paper (the added-value of spectral estimates), more details should be provided with a more precise view on the state of the art regarding the modelling of wave-current interaction in 3D (cf. my general comment above).*

We acknowledge the importance of providing a comprehensive overview of the state of the art of the modelling of wave-current interaction in 3D. In response to the Reviewer's valuable suggestion, we will enhance the end of the introduction section to ensure a more thorough and detailed presentation of the current state of the field. By incorporating the references suggested by the Reviewer in his/her general comment, we will be able to enhance the comprehensiveness of our manuscript.

8) *Line 46: I do not think that "initiated" is correct, given that most of the work come from that of Uchiyama et al. (2009, 2010). The study by Marchesiello et al. (2015) brings no new development on the wave-current interaction part, and the authors do not even mention the CROCO modelling system.*

We agree with the Reviewer and will revise the sentence in question to better reflect the contributions of the cited works.

9) *Line 52: Again, what do the authors mean? This is literally the model developed in Uchiyama et al. (2010) and used in CROCO.*

See above point. This part of the introduction will be modified to better reflect the contributions of the cited works.

10) *Lines 52-55: If the model used in Marchesiello et al. (2015) is so accurate, why not using their configuration to test the added-value of the new developments in the context of the nearshore region?*

See the response to the third major point.

APPLICATION SITE AND DATA

11) *This section feels quite tedious to read, with too many and sometimes irrelevant details given the scope of the present study. I think that the authors need to reshape this section and keep the most relevant information only.*

We agree with the Reviewer that, despite well in line with the funding project of our research work, the description of the sedimentary environment in the Bay of Somme does not include relevant information for the scope of the present study. We will then remove from the section the paragraph describing sediment transport, seabed features and infilling sedimentary process characterising the bay.

12) *Line 86: "[. . . ] a not fully developed wind sea. The energetic swell are generated*

*on limited fetches with a maximum length of 400 km". What do the authors mean? Swell developing over 400 km-long fetches?*

Yes, the Reviewer is right.

13) *Lines 102-107: a careful read is the minimum to provide before submission.*

Please, see our response to the general comment.

14) *Line 110: "2 meters above the bottom", so not over the whole water column as stated?*

The Reviewer is right. The text will be corrected.

METHODS

15) *Description of WAVEWATCH-III: to my understanding, this model solves the evolution of directional wave spectra in the wavenumber space. Why is this equation different from that given in the model manual?*

The equation is identical to the one found in the WAVEWATCH-III manual, with the only distinction being that it is expressed in terms of wave energy and frequency, as opposed to wave action.

16) *Line 140-141: what wetting and drying means for spectral wave modelling.*

Wetting and drying in the spectral wave model operates in a manner consistent with the hydrodynamic model. It refrains from computation in grid cells where the instantaneous water depth falls below a specified threshold value.

17) *There seems to be a typo in Tm01 in the manuscript, even the schematic of Fig. 2.*

The typo will be corrected throughout the text and in the schematic of Figure 2.

18) *Line 287: similar to earlier comment, what do the authors mean here? The wave setup or set-down are not forcing terms but result from the wave forces.*

The text ill be corrected.

NUMERICAL EXPERIMENTS

19) *Why using such a small scale model, which depend so strongly on wave and tidal forcing used at the boundaries? In particular, the tidal forcing is derived from a 2D hydrodynamic model, how does that affect the vertical shear observed at the instrument positions? The discussion of the results presented in the paper depends too heavily on this aspect for being ignored and not investigated.*

We agree with the Reviewer that our discussion of the results depends very much on the vertical shear observed at the instrument position to do not investigate the control exerted by the imposed tidal forcing on the results. As discussed in detail in the response to the second major comment, we conducted an additional numerical simulation with an extended computational domain, placing the measurement sites further from the boundaries where we imposed tidal

levels and currents. These supplementary results indicated the independence of our previous results from the relative distance between the forcing boundary and the measurement locations.

20) *Lines 310-311: Why such drastic choices? Can this model be reliably used in a nearshore context? Or is it suffering from instabilities? In any case, given this information, the authors simply cannot discuss shallow water processes like wave setup.*

We did not account for wave effects on currents below 2 meters water depth because the interpolation of the low-resolution bathymetric data resulted in steep seabed gradients within the bay, leading to numerical instabilities in the surf zone. We have recently collected accurate shallow-water high-resolution bathymetric data within the bay. In the forthcoming phases of our modelling of the Bay of Somme, we will use these new data and test the reliability of our model in the nearshore context.

21) *The impact of surface waves on the wind drag coefficient is not described. Have the authors investigated this? How did they make sure that their tests are all consistent with this regard?*

We appreciate the Reviewer's consideration of the potential impact of the model domain's scale on the representation of wind contributions. While it is true that the wind contribution near the boundary may be subject to some uncertainty due to the reduced extension of the model domain, we would like to emphasise that the identified tendency observed in our results remains valid. The key point is that the identified tendency, specifically the acceleration of the current profile when accounting for additional spectral wave field exchanges, is not contingent on the specific uncertainties related to wind contributions. Even though some uncertainty may exist due to the scale of the model domain and its influence on the reproduction of a realistic 3D profile, the observed tendency is consistent and supported by the data.

RESULTS AND DISCUSSIONS

22) *Line 331: why is "fresh" relevant here? I might be missing some specific term here.*

Here we are referring to the Beaufort wind scale, where fresh to strong breeze are classified depending on the wind speed.

23) *Lines 333: Why switching to Hrms, is that for reducing the errors of the model? Though the authors consider their model "validated" (line 337), at the peak of the event they consider, half the wave energy is missing in the model. In my opinion, this is far from being satisfactory, given that the authors then discuss added-value of the order of 1 cm/s in terms of currents, without proper sensitivity analysis of other relevant parameters (wind drag coefficient, mixing length, amount of TKE injected in the water column and so on).*

We used both significant wave height and its root-mean-square value throughout the manuscript due to the fact that the wave model provides results in terms of significant wave height, while the hydrodynamic model gives root-mean-square values. We acknowledge the Reviewer's concern regarding the discrepancy in wave energy representation during the peak of the discussed event.

It is important to note that this limitation does not arise from the specific model developments presented in our study but is associated with the spectral wave model used to provide boundary conditions with full spectra, a computationally demanding yet best practice in wave modelling. Furthermore, it's worth emphasizing that the representation of tidal currents is much more accurate than that of waves in our model. The primary focus of our study is to assess the added value of the spectral terms introduced in the representation of wind and wave effects on macrotidal currents, which can still be meaningfully discussed with the current model configuration. We observe that the predicted current values, when considering these additional terms, tend to be closer to the measurements, resulting in differences of approximately 10% of the current magnitude (2-3 cm/s in 20-30 cm/s currents, see bottom left panel of Figure 13). Nonetheless, we agree with the Reviewer that addressing the issue of wave energy representation in future modeling efforts is important. We appreciate the Reviewer's feedback, which will guide our ongoing work to further enhance the accuracy and reliability of our model, especially regarding wave-current interactions

24) *Line 343: What is a wavenumber velocity?*

This part of the text will be removed.

25) *Line 358: this statement is not supported by the material presented in the manuscript.*

This part of the text will be removed.

26) *Line 360-363: To me, it simply shows how dependent the local model is to the forcing, and that there is a systematic low bias at every high tide (half a meter). Then, the authors provide some explanation to the observed phase shift, while it clearly comes from the forcing. How does the original hydrodynamic model compare with the data?*

Please see the response to the second major point.

27) *Line 382-383: Not really true since a better match is systematically obtained with the previous configuration during ebbs.*

True, but a better match is systematically obtained with the coupled wind-wave-current configuration during the remaining phases of the tidal cycle, resulting in the same R2 and RMSE values on which this sentence is based.

28) *Line 390: "[. . . ] due to the presence of sub-grid scale bedforms". Again not supported. Please check the forcing.*

The sentence will be modified without referring to the presence of sub-grid scale bedforms. Regarding the forcing, as discussed in our response to the second major point, we checked the forcing and we assessed that it does not overly control the obtained results.

ASSESSMENT OF VERTICAL CURRENT PROFILES FOR CONTRASTING EVENTS

29) *Line 408: How is wind drag coefficient computed, and how did the authors make sure all simulations are consistent with this regard? i.e. is the wind contribution similar when waves are accounted for?*

The wind drag coefficient is computed according to the formulation of Smith (1988), which estimates the coefficients for sea surface wind stress as a function of wind speed while accounting for a constant Charnock coefficient. In the present study, the latter is computed assuming fetch-limited coonditions. In the discussion of the results, our aim was to methodically isolate the distinct contributions stemming from various forcing factors. To achieve this, we conducted a series of simulations in which we incrementally introduced different metocean forcings while ensuring that their effects remained as isolated as possible. As part of this approach, we deliberately did not consider the influence of waves on winds through adjustments to the Charnock coefficient. In other words, we kept the wind contribution consistent, irrespective of whether waves were factored into the model. This deliberate choice allowed us to scrutinise and quantify the impact of other variables while minimising the confounding effects of altered wind conditions due to wave effects and viceversa.

30) *Lines 413-414: What about the impact of the mixing length and its parameterisation (end of Section 3.3.5) on the current magnitude at the top of the water column?*

We did not perform sensitivity tests on the impact of the mixing length parameterisation employed. In Section 3.3.5 we introduced the turbulence model to highlight the difference in its choice with respect to former versions of ROMS, which include some of the additional spectral wave terms discussed in our study while using a different turbulence model.

31) *Line 433: I rather see a 1 and 3 cm/s increase depending on the configuration.*

The Reviewer is right. We will modified the text accordingly.

32) *Line 435: "[. . . ] improves the accuracy of the results". Given the results, this is all relative, i.e. whether taken at the surface or bottom. What are the rmsd for depth-averaged values and how do they compare?*

The results using the new spectral field exchange show an improved matching with the measurements both close to the free-surface and the bottom. Also, since the entire paper is focused on the 3D modelling of wave-current interactions, we do not see the point in computing stats associate to depth-averaged values.

33) *Line 437: Why jumping over 3 Figures? Also, this specific graph shows velocities (m/s) and heights (m) over the same axis, a different one should be used.*

The figure will be modified according to the Reviewer's comment.

34) *Line 447: Given that the boundary remains close to the measurements location, too much uncertainty exists on whether or not the model develops a realistic 3D profile (i.e. taking into account the response to winds etc). In this context, the statement is speculative and should be investigated further.*

We removed te speculative sentence from the text.

35) *Line 443-444: What do the authors mean here? Please develop.*

Probably there is a typo in the line numbers. At these lines, we just observed how wind aligned

with tidal propagation accelerates further the free-surface velocity.

36) *Line 451-452: So why choosing this limited experimental data set or focusing on the upper section of the water column?*

To understand why we focus on this particular data set, please see the response to the third major point. It's crucial to emphasise that the primary focus of our paper is to evaluate the added-value of incorporating additional spectral wave terms in the computation of wave-current interactions, rather than concentrating on the details of turbulence modelling in these interactions. We employed a standard turbulence model with standard boundary conditions to ensure a fair and consistent comparison between simulations with and without the spectral terms.

37) *Line 465: Not supported. Using more carefully designed numerical experiments, including synthetic cases would be extremely useful here.*

The sentence will be removed from the text.

38) *Line 472-473: Here, I think it is misleading to compare the wave breaking terms for bulk and full spectral quantities. The bulk representation has so far only been derived and used for the case of depth-induced wave breaking (Uchiyama et al., 2010; Kumar et al., 2012) and not for whitecapping, which is the dominant dissipative process for waves in the present case. Whitecapping affects mostly high-frequency components, so the comparison is not relevant in my opinion. This made me realise that the authors did not describe the full spectral representation of the wave forces (not that of dissipation).*

We agree with the Reviewer that directly comparing these terms may not be suitable, given that the bulk representation has primarily been applied in the context of depth-induced wave breaking rather than whitecapping, which is the dominant dissipative process for waves in our study area. However, we would like to clarify that our intention was not to compare bulk and full spectral representations of wave breaking. Instead, our objective was to assess the improvement resulting from our model developments, which include the representation of whitecapping dissipation in addition to depth-induced breaking. Since no depth-induced breaking occurs in the measuring location where the data discussed in this study have been collected, the observed changes in the solution are primarily due to the consideration of whitecapping effects. To make this point clearer, we will revise the corresponding text to better express the purpose of this comparison.

39) *Line 490: This is all relative, we speak of 0.5 cm/s differences while wave model performances are poor and RMSE for currents is 10 cm/s, with much larger errors locally.*

Please consider that the disparities in current magnitude are, in certain instances, greater than 3 cm/s. Also, when addressing variations slightly below 1 cm/s in a Stokes drift that fluctuates between 10 and 5 cm/s, this accounts for approximately 10% of the overall value.

40) *Fig. 17: Please clarify the representation of water levels as it is not clear. Also, in some Figures, the vertical datum for depth is not clear.*

Figure 17 will be changed to clarify the representation of water levels.

41) *Line 497: Is that so? Unless there are inconsistencies in ROMS/CROCO, the peak wavenumber used in the monochromatic formulation should be Doppler-shifted too in full-coupled configurations.*

The Reviewer is right, and we will remove the sentence from the manuscript. Initially, this section aimed to elucidate the distinctions between one-way and two-way coupled simulation outcomes. However, we later decided to exclude the one-way simulations from the paper to enhance conciseness.

42) *Line 505: Please be consistent. In some sections of the manuscript, refraction processes are well represented and in this one, they are likely misrepresented. In both cases, the analysis is poorly supported by results.*

Since we will removed from the text the parts dealing with wave processes in shallo waters, the depth-induced refraction process will be described only as likely misrepresented. We will deal with this issue using higher resolution nested models in shallow waters in our future modelling works.

43) *Line 506: So why not increasing the spatial resolution of the model? And why analysing processes such as depth-induced refraction and breaking?*

Please see the response to the third major comment.

44) *Line 515-520: How is the surf zone identified here? Does that include region where whitecapping occurs? Like many sections and physical interpretations of results, this discussion is poorly supported.*

We will modified the text replacing the use of the term 'surf zone' to avoid any confusion.

45) *Line 544-545: I struggle to understand and believe the key message here.*

The observed slight phase shift in tidal floods between modelled and measured currents does not depend on random metocean forcing. As it is not related with wind and wave forcing, it does not impede our ability to assess the effects of wind and wave forcing on macro-tidal currents, which is the primary focus of our study. Thus, this phase shift is not a hindrance to the evaluation of our model developments related to wave and wind influences on currents.

[Figure]

Figure 1: **New model results in the extended computational domain**. Depth-averaged current magnitudes (coloured shading) with superimposed current vectors (black arrows) computed from the new simulation across its whole domain at the peak of the flood tidal flow the 2 August 2008 at 11 pm. White lines are bathymetric contours while black circles are the extracting (ADCP and AQDP) locations.

[Figure]

Figure 2: **New model results in the previous computational domain**. Depth-averaged current magnitudes (coloured shading) with superimposed current vectors (black arrows) computed from the new simulation across the old domain at the peak of the flood tidal flow the 2 August 2008 at 11 pm. White lines are bathymetric contours while black circles are the extracting (ADCP and AQDP) locations.

[Figure]

Figure 3: **Time series of newly computed tidal currents versus AQDP data**. Model results (lines) are compared with AQDP (dots) measurements in terms of (top) current magnitude and (bottom) 'going to' direction at 1 m above bottom (mab).

[Figure]

Figure 4: **Vertical profiles of newly computed tidal currents versus ADCP data**. Measured (squares) and modelled (continuous lines) vertical profiles of the current on 2 August 2008 at 2.30 pm during tidal reversal (left panel). Measured (squares) and modelled (continuous lines) vertical profiles of the current on 2 August 2008 at 11.00 pm during tidal ebb (right panel).

---

## Author Response (AR1)

**Reply to Reviewer 1**

**General comment**

*This paper presents a new coupling procedure between the 3D ocean circulation model CROCO and the spectral wave model WAVEWATCH-III, handled through the OASIS-MCT coupler, in order to improve the representation of wave-current interactions in the coastal region. More precisely, efforts are made to incorporate wave quantities derived from the full (bi-dimensional) wave spectrum that better represents "real" directionally and frequency-broad wave fields. This is in contrast with "monochromatic" approximations that use a representative wavenumber (e.g. mean or peak) as originally used and implemented in ROMS (Uchiyama et al., 2009, 2010).*

*After a broad overview of the different approaches for simulating wave-current interaction in the coastal context, the paper describes the new forcing terms. The rest of the paper focuses on a series of numerical experiments performed in the geographical setting of the Bay of Somme, France, a macrotidal site located in the English Channel. In situ data (vertical current profiles at two locations) are used to assess the model performances as well as to evaluate the added value of the newly-proposed coupling procedure.*

*I found the manuscript relatively well organised, though not particularly well written. At least, it could have used a few more readings by the authors: e.g. some internal notes made by the authors remain in the core of the manuscript (see at lines 102-107), which is not acceptable in my opinion. The efforts to incorporate full spectral representation of wave quantities are welcome, as it should help to get a better and more realistic representation of wave forces while modelling wave-current interactions with this modelling system. The (scientific) novelty is not obvious, though, since similar efforts made by other authors in ROMS are overlooked, and such spectral representation already exists in other modelling systems. So it is not clear to me whether this contribution justifies a publication or not. I have several major comments on the present work that, I think, should be addressed before a resubmission.*

We express our sincere appreciation to the Reviewer for conducting a thorough review of our manuscript and for providing valuable feedback that has greatly contributed to the enhancement of our work. We fully understand the importance of addressing the major comments raised by the Reviewer before resubmission.

In terms of the organisation and clarity of the manuscript, we acknowledge the Reviewer's valid point about the need for further editing and refinement of the text. We would like to clarify that the issue raised in lines 102-107 occurred during a minor revision phase, during which we were specifically instructed to change only the title and availability statement. Regrettably, an erroneous copy-paste during the final submission inadvertently included internal notes from another manuscript. We deeply regret this oversight. However, we want to assure the Reviewer that, apart from this isolated issue, the manuscript underwent multiple reviews and revisions by the authors prior to the initial submission to ensure its quality and coherence.

We concur with the Reviewer that certain relevant prior work was initially overlooked. Building upon the Reviewer's citations, we have incorporated references to pertinent previous studies and emphasised the specific contributions of our study in light of these works, particularly in terms of our utilisation of full spectral representation of wave quantities. We also recognise the importance of highlighting the unique aspects of our work. To achieve this, we have revised the introduction section to better underscore the novelty and distinctiveness of our study within the broader context of wave-current interaction modelling, discussing how our approach differs from previous efforts, both within our modeling system and in other modeling systems. We firmly believe that our study provides valuable insights and advancements in the field of wave-current interaction modelling, and that the revised manuscript conveys this effectively.

Below detailed point-by-point responses to all referee's comments are provided and all related changes in the revised manuscript are specified. Line numbering refers to the track-change version of the revised manuscript.

**Major comments**

1) *The manuscript completely ignores many relevant and recent works that have looked into these aspects, some within the frame of the same modelling system. Within ROMS, from which CROCO is derived, the works by L. Romero and the UCLA team are completely overlooked (in particular Romero et al., 2021; Hypolite et al., 2022). This also includes the work initiated by N. Kumar (Kumar et al., 2017; Liu et al., 2021) that has evaluated the impact of the definition used for the Stokes drift velocities on the modelling of wave-current interactions at regional scales. The present manuscript also ignores modelling systems already incorporating full spectral wave quantities such as MOHID (Delpey et al., 2014) or SCHISM, which has been used to simulate and analyse the wave-induced nearshore circulation in numerous realistic settings at both regional (Guérin et al., 2017; Lavaud et al., 2020; Pezerat et al., 2022) and local (Martins et al., 2022) scales.*

We highly appreciate the valuable comment provided by the Reviewer, which has guided us in improving the quality of our manuscript with the refinement of the referenced literature and the introduction of our model developments. In response to the Reviewer's observations, we have addressed the omission of references to previous model developments within the same modelling framework and those from other model systems. Of particular significance is the inclusion of reference to the work initiated by Kumar et al. (2017). These references have been thoughtfully incorporated into the introduction of our manuscript to provide a more comprehensive context for our research.

To effectively underscore the unique contributions of our manuscript in relation to the works mentioned by the reviewer, we provide a detailed discussion of each specific work, highlighting both connections and distinctions with our study.

**Works Within the Same Modelling System**

*Kumar et al. (2017)* proposed a significant advancement in modelling wave-current interactions through the advocacy of spectral reconstruction for the forcing and coupling of wave-current models. This approach, rooted in the partitioning of WaveWatchIII, delivers more accurate estimates of the Stokes drift when compared to the spectral peak monochromatic approximation of *Uchiyama et al. (2010)*, which often results in underestimations. Later, *Liu et al. (2021)* confirmed the need of spectral estimates of the Stokes drift for both deep- and intermediate-water applications. Our study aligns with this methodology by employing full spectra from a regional spectral model to force our wave model and incorporating spectral estimates of the Stokes drift. Moreover, we have introduced additional spectral field exchanges to further refine the description of wave-current interactions, thereby distinguishing our novel approach.

*Romero et al. (2021)* introduced a novel ROMS WEC (Wave Effects on Currents) framework that extended the monochromatic approach of *Uchiyama et al. (2010)*. This extension encompassed spectral approximations of Stokes drift, Bernoulli head, and wave-induced vertical mixing, achieved through iterative computations and a switching mechanism between deep- and shallow-water formulations. *Hypolite et al. (2021)* used this framework to explore the impacts of the spectral approach on ocean mesoscale circulation variability, revealing relatively modest wave effects despite larger Stokes drift estimates. In consonance with their work, we have incorporated the exchange of spectral Bernoulli head in our model coupling. Additionally, our enhancements encompass novel spectral field exchanges designed to provide a more precise representation of the wave-induced bottom boundary layer. This includes the exchange of spectral bottom wave orbital velocity, which we have demonstrated to have a comparable impact on macro-tidal current profiles to that of the Stokes drift in our specific application. Notably, our study accentuates the significant influence of wave effects on macro-tidal currents, further distinguishing our research from prior investigations.

**Works on Different Model Systems Incorporating Full Spectral Wave Quantities**

While the references cited by the Reviewer describe developments akin to ours, all aimed at achieving a more comprehensive description of wave-current interactions, our study delves into aspects specific to the employed modelling system, setting it apart from the frameworks mentioned by the Reviewer in several key ways.

Regarding technical aspects, our coupled model uses different hydrodynamic and wave spectral models than the cited systems. CROCO and WWIII are not integrated with each other. To do so, the CROCO coupled system uses the OASIS-MCT coupler, which is a set of libraries allowing for parallel exchanges and grid interpolations between different models. This framework has multiple advantages such as the possibility of choosing different grids for the different models (and eventually different nesting strategies), the possibility of choosing the coupling frequency, the exchanged variables, etc. This interface is thus particularly flexible and allows coupling CROCO with any model included in the OASIS-MCT library. Furthermore, the computational grids used by CROCO are structured rather than unstructured as the cited modelling systems.

Regarding historic aspects, ROMS was originally designed for regional oceanic applications. CROCO inherited from ROMS and recent model developments added up new capacities to link the regional, coastal and nearshore scales. Our work, in line with these recent advances, concentrates on its adaptation for intermediate-water macro-tidal conditions, distinguishing it from the cited modelling frameworks which are more nearshore-oriented.

To refine the referenced literature and better introduce our model developments in light of these existing works, we have added the text at lines 74-88 which provides a more comprehensive context for our research within the ROMS modelling system, and we have also added an entire paragraph (lines 89-102) to discuss similar model developments already existing in other modelling systems.

2) *The present numerical setup is far from being ideal to discuss the added value of (full) spectral representation of wave quantities, for several reasons. The hydrodynamics simulated here is overly controlled by the forcing used at boundaries, both in terms of water levels, currents and waves, to the point that the reader asks himself what is the added value of using a local model. For instance, the wave results at the measurement site show no improvements over the large scale model used as forcing. Furthermore, water levels and currents are (to my understanding) forced from a 2D hydrodynamic model. Considering the importance given to the vertical shear, how can we be sure that the boundary is taken sufficiently far away from the location where measurements were obtained and vertical velocity profiles discussed? I suspect that the boundary is too close to the measurement stations in order to discuss differences of the order of 1 cm/s at the top layer of the water column.*

We duly acknowledge the Reviewer's concern regarding the influence of wave boundary conditions on our simulation. It is indeed true that the hindcast data used to force the wave model control the hydrodynamics while exhibit some biases in our study area. Nevertheless, it is crucial to emphasize that the use of full spectra at the boundaries was made because is the best (and most computationally demanding) practice in modelling spectral waves, particularly when investigating complex phenomena such as spectral Stokes drift and wave-current interactions (as highlighted in Kumar et al., 2017, and Liu et al., 2021, both now cited in the revised manuscript).

Addressing the concern related to water levels and currents, we acknowledge the possibility that the proximity of the southern boundary to the measurement sites could result in an appearance of excessive control by the imposed forcing. To investigate this, we conducted an additional numerical simulation with an extended computational grid in the south and west directions, effectively placing the measurement sites further from the open boundaries (see Figure 1).

The results of this supplementary simulation align closely with the outputs of the original simulation which solely employed tidal forcing. This congruence is evident in terms of circulation patterns (as illustrated in the tidal flood snapshot, Figure 2, which can be compared with Figure 6 of the submitted manuscript), time series of near-bottom currents (Figure 3), and discussed vertical profiles (Figure 4).

These additional results convincingly demonstrate that the boundary was situated at a sufficient distance from the measurement locations where vertical velocity profiles were examined. It is vital to emphasise also that, when comparing velocity vertical profiles computed considering only tidal forcing prescribed at different open boundaries (Figure 4), the observed discrepancies between original (black lines) and extended (green lines) setups are one order of magnitude lower than those found comparing vertical profile influenced by winds and waves (see panel C of Figure 11 in the track-change version of the manuscript).

Based on these additional results, we assert with confidence that the choice of tidal forcing in our simulations does not exert undue control over the modelled hydrodynamics. This support our validation of the macro-tidal currents influenced by wind and waves in the bay of Somme and the discussion of the substantial added value derived from the incorporation of additional spectral wave field exchanges. We have mentioned the results of this additional simulation in the section dedicated to the setup of the numerical experiments (lines 355-360) to strengthen our case.

3) *The discussion of wave processes, particularly in shallow water depths, is in my opinion weak throughout the paper. Discussion on depth-induced refraction (short waves in relatively deep water depth) is not supported by the numerical results, while the section dealing with wave breaking in shallow water (set-down and setup) is really weak, to say the least. Reading that wave effects are "removed" in depths less than 2 m (why is that so?), how can we then expect a wave setup to form near the shoreline? The influence of using a spectral representation is expected to be much more important in wave-dominated shallow-water environments, so we expect this study to also consider such environments where this model has already been applied (e.g. Duck N.C. in Uchiyama et al., 2010; or the Biscarosse site in Marchesiello et al., 2015).*

We concur with the Reviewer's perspective that the use of a spectral representation is more significant in wave-dominated shallow waters. In such environments, the effects of waves on currents can manifest with particular prominence, a factor that is less pronounced in intermediate water depths, as the locations of the measurements discussed in this study.

To provide a comprehensive context for our study and elucidate the rationale for our choice of modelling application, it is essential to place our research within the framework of its funding project. The DEMLIT project, funded by SHOM in collaboration with the University of Caen Normandy, is dedicated to investigate the sand filling dynamics in the Bay of Somme. The submitted manuscript represents the initial phase of this project, focusing on the macro-tidal circulation off the bay and its interactions with offshore winds and waves. To do this we employed available tidal atlases and low-resolution bathymetric data from SHOM complemented by in-situ measurements from the University. It is important to note that we did not account for wave effects on currents below 2 meters water depth, as the interpolation of the low-resolution bathymetic data within the bay resulted in steep seabed gradients, leading to numerical instabilities in the surf zone and not allowing a proper representation of the nearshore.

This model will serve as a parent model, providing a foundation for higher-resolution, shallower-water nested applications. These forthcoming applications will explore hydrodynamics and sediment transport processes within the bay. Notably, we have collected new shallow-water, high-resolution bathymetric data and ADCP measurements in the surf zone to inform and validate this forthcoming phase of the modelling.

We understand the significance of accurately reproducing wave processes in shallow waters, especially while striving to improve the representation of wave effects on currents, and we acknowledge that previous versions of our model have been applied in such settings, as highlighted by Uchiyama et al. (2010) and Marchesiello et al. (2015). However, it is crucial to underscore the primary objective in our current study, which is centred on model the interactions of macro-tidal currents with offshore winds and waves. Given this clarification, we have removed from the section dealing with the assessment of modelled wave the last paragraph discussing wave breaking in shallow waters (lines 401-417). These aspects, though undeniably important, will be reserved for the future extensions of our modelling efforts.

4) *Instead of a lengthy description of the field site, and given the chosen journal, we expect much more description on the coupling procedure: where there files modified in the wave or hydrodynamic models or can new versions of these models be pulled directly? Can the new coupling procedure be reproduced elsewhere easily? What about interpolation of the different fields and where is it handled?*

We appreciate the Reviewer's feedback and agree that a more detailed description of the coupling procedure is warranted.

All the additional spectral fields required for the proposed advanced coupling were already incorporated into the WAVEWATCH-III model, and we simply added the necessary exchanges in CROCO to compute the wave forcing terms. These modifications have been available in CROCO since the v1.2 release. Both the 'monochromatic' and 'full' coupling procedures are accessible for comparison in different configurations, and users can easily switch between them by activating or deactivating a pre-processing key.

The OASIS coupler handles all data exchanges, including interpolations on different grids. Coupling information is specified in a parameter file exclusively used by OASIS. Therefore, applying this new coupling to another configuration and assessing the impact of our developments should not pose significant challenges.

These additional details have been added to the section dealing with the description of the coupling procedure at lines 331-333 and lines 336-340.

**More specific comments:**

ABSTRACT

1) *Lines 7-8: Given the presented results, and the comments above, I think that the "sig-*

*nificant" currently needs to be toned down. Considering nearshore situation with adequate numerical experiments could surely better highlight the impact of using (full) spectral wave quantities.*

Following the Reviewer's suggestion, we have carefully reconsidered the wording in the abstract to accurately reflect the main outcomes of our study while taking into account its limitations in terms of nearshore modelling and the primary scope of our work.

2) *Lines 15-16: What about nearshore environments, where Bernoulli heads and breaking acceleration terms will be the dominant forcing of the nearshore circulation?*

In response to the Reviewer's valuable feedback, we have carefully revised the text throughout the entire manuscript to eliminate any ambiguity regarding the study's objective. We have explicitly narrowed down the study objective from 'modelling wave-current interactions in coastal areas' to 'modelling interactions of macro-tidal currents with winds and waves'. This adjustment now ensures greater clarity and alignment with the primary goals of our research.

INTRODUCTION

3) *Line 21: "Wave-induced currents and setup" are not forcing terms.*
The text have been corrected.

4) *Line 26: Reformulate with "wave-dominated environments"?*
As we clarified in our response to the third major comment, the specific focus of our study does not revolve around 'wave-dominated environments'. We have taken extensive measures to overhaul the entire introduction section, ensuring that the text provides a clear and accurate representation of the paper's objective.

5) *Lines 34-35: "Three-dimensional modelling of the wave-induced flow was requested". I think that the authors can find much better context and needs for resolving 3D currents in the nearshore context (e.g. the transport of sediment, particles or tracers).*
The sentence have been modified in accordance with the primary objective of our study. We appreciate the Reviewer's suggestion, and it's essential to note that even in intermediate waters within macro-tidal environments, the resolution of the vertical current profile is crucial to understand and model the transport of sediment, particles, and tracers. While the effects of winds and waves may be comparatively less pronounced, the accuracy of the vertical current profile remains instrumental in capturing these important aspects even in the discussed intermediate-water setting.

6) *Line 35: Specify what these theories are used for.*
The text have been updated to specify that these theories derive depth-dependent expressions of the wave-averaged flux of momentum due to waves to study the interaction of currents and waves in water of finite depth.

7) *Line 40: "With simplifications". The end of this section feels too light, and considering*

*the scope of the paper (the added-value of spectral estimates), more details should be provided with a more precise view on the state of the art regarding the modelling of wave-current interaction in 3D (cf. my general comment above).*

We acknowledge the importance of providing a comprehensive overview of the state of the art of the modelling of wave-current interaction in 3D. In response to the Reviewer's valuable suggestion, we have enhanced the end of the introduction section to ensure a more thorough and detailed presentation of the current state of the field. By incorporating the references suggested by the Reviewer in his/her general comment, we have been able to enhance the comprehensiveness of our manuscript.

8) *Line 46: I do not think that "initiated" is correct, given that most of the work come from that of Uchiyama et al. (2009, 2010). The study by Marchesiello et al. (2015) brings no new development on the wave-current interaction part, and the authors do not even mention the CROCO modelling system.*

We agree with the Reviewer and we have revised the sentence in question to better reflect the contributions of the cited works.

9) *Line 52: Again, what do the authors mean? This is literally the model developed in Uchiyama et al. (2010) and used in CROCO.*

See above point. This part of the introduction has been modified to better reflect the contributions of the cited works.

10) *Lines 52-55: If the model used in Marchesiello et al. (2015) is so accurate, why not using their configuration to test the added-value of the new developments in the context of the nearshore region?*

Please, see the response to the third major point.

APPLICATION SITE AND DATA

11) *This section feels quite tedious to read, with too many and sometimes irrelevant details given the scope of the present study. I think that the authors need to reshape this section and keep the most relevant information only.*

We agree with the Reviewer that, despite well in line with the funding project of our research work, the description of the sedimentary environment in the Bay of Somme does not include relevant information for the scope of the present study. We have then removed from the section the paragraph describing sediment transport, seabed features and infilling sedimentary process characterising the bay.

12) *Line 86: "[. . . ] a not fully developed wind sea. The energetic swell are generated on limited fetches with a maximum length of 400 km". What do the authors mean? Swell developing over 400 km-long fetches?*

Yes, the Reviewer is right. The sentence has been rephrased.

13) *Lines 102-107: a careful read is the minimum to provide before submission.*

Please, see our response to the general comment.

14) *Line 110: "2 meters above the bottom", so not over the whole water column as stated?*

The Reviewer is right. The text have been corrected.

METHODS

15) *Description of WAVEWATCH-III: to my understanding, this model solves the evolution of directional wave spectra in the wavenumber space. Why is this equation different from that given in the model manual?*

We have decided to replace the equation in question with the original one found in the WWIII manual to avoid any potential confusion.

16) *Line 140-141: what wetting and drying means for spectral wave modelling.*

Wetting and drying in the spectral wave model operates in a manner consistent with the hydrodynamic model. It refrains from computation in grid cells where the instantaneous water depth falls below a specified threshold value. However, to avoid any potential confusion, we have decided to remove this information from the text.

17) *There seems to be a typo in Tm01 in the manuscript, even the schematic of Fig. 2.*

The typos have been corrected throughout the text and in the schematic of Figure 2.

18) *Line 287: similar to earlier comment, what do the authors mean here? The wave setup or set-down are not forcing terms but result from the wave forces.*

The text has been corrected.

NUMERICAL EXPERIMENTS

19) *Why using such a small scale model, which depend so strongly on wave and tidal forcing used at the boundaries? In particular, the tidal forcing is derived from a 2D hydrodynamic model, how does that affect the vertical shear observed at the instrument positions? The discussion of the results presented in the paper depends too heavily on this aspect for being ignored and not investigated.*

As discussed in detail in the response to the second major comment, we conducted an additional numerical simulation with an extended computational domain, placing the measurement sites further from the boundaries where we imposed tidal levels and currents. These supplementary results prove the independence of our previous results from the relative distance between the forcing boundary and the measurement locations.

20) *Lines 310-311: Why such drastic choices? Can this model be reliably used in a nearshore context? Or is it suffering from instabilities? In any case, given this information, the authors simply cannot discuss shallow water processes like wave setup.*

We did not account for wave effects on currents below 2 meters water depth because the interpolation of the low-resolution bathymetric data resulted in steep seabed gradients within the bay, leading to numerical instabilities in the surf zone. We have recently collected accurate shallow-water high-resolution bathymetric data within the bay. In the forthcoming phases of our modelling of the Bay of Somme, we will use these new data and test the reliability of our developments using a higher resolution application in the nearshore context.

21) *The impact of surface waves on the wind drag coefficient is not described. Have the authors investigated this? How did they make sure that their tests are all consistent with this regard?*

The wind drag coefficient has been computed according to the formulation of Smith (1988), which estimates the coefficients for sea surface wind stress as a function of wind speed while accounting for a constant Charnock coefficient. We did not consider the influence of waves on winds through adjustments to the Charnock coefficient. Please, see also comment to comment 29.

RESULTS AND DISCUSSIONS

22) *Line 331: why is "fresh" relevant here? I might be missing some specific term here.*

Here we are referring to the Beaufort wind scale, where fresh to strong breeze are classified depending on the wind speed.

23) *Lines 333: Why switching to Hrms, is that for reducing the errors of the model? Though the authors consider their model "validated" (line 337), at the peak of the event they consider, half the wave energy is missing in the model. In my opinion, this is far from being satisfactory, given that the authors then discuss added-value of the order of 1 cm/s in terms of currents, without proper sensitivity analysis of other relevant parameters (wind drag coefficient, mixing length, amount of TKE injected in the water column and so on).*

We used both significant wave height and its root-mean-square value throughout the manuscript due to the fact that the wave model provides results in terms of significant wave height, while the hydrodynamic model gives root-mean-square values. We acknowledge the Reviewer's concern regarding the discrepancy in wave energy representation during the peak of the discussed event. It is important to note that this limitation does not arise from the specific model developments presented in our study but is associated with the spectral wave model used to provide boundary conditions with full spectra, a computationally demanding yet best practice in wave modelling. Furthermore, it's worth emphasising that the representation of tidal currents is much more accurate than that of waves in our model. The primary focus of our study is to assess the added value of the spectral terms introduced in the representation of wind and wave effects on macrotidal currents, which can still be meaningfully discussed with the current model configuration. We observe that the predicted current values, when considering these additional terms, tend to be closer to the measurements, resulting in differences of approximately 10% of the current magnitude (2-3 cm/s in 20-30 cm/s currents, see bottom left panel of Figure 13). Nonetheless,

we agree with the Reviewer that addressing the issue of wave energy representation in future modelling efforts is important. We appreciate the Reviewer's feedback, which will guide our ongoing work to further enhance the accuracy and reliability of our model, especially regarding wave-current interactions

24) *Line 343: What is a wavenumber velocity?*

This part of the text has been removed.

25) *Line 358: this statement is not supported by the material presented in the manuscript.*

This part of the text has been removed.

26) *Line 360-363: To me, it simply shows how dependent the local model is to the forcing, and that there is a systematic low bias at every high tide (half a meter). Then, the authors provide some explanation to the observed phase shift, while it clearly comes from the forcing. How does the original hydrodynamic model compare with the data?*

Please, see the response to the second major point.

27) *Line 382-383: Not really true since a better match is systematically obtained with the previous configuration during ebbs.*

True, but a better match is systematically obtained with the coupled wind-wave-current configuration during the remaining phases of the tidal cycle, resulting in the same R2 and RMSE values on which this sentence is based.

28) *Line 390: "[. . . ] due to the presence of sub-grid scale bedforms". Again not supported. Please check the forcing.*

The sentence has been modified without referring to the presence of sub-grid scale bedforms. Regarding the forcing, as discussed in our response to the second major point, we checked the tidal forcing and we assessed that it does not overly control the obtained results.

ASSESSMENT OF VERTICAL CURRENT PROFILES FOR CONTRASTING EVENTS

29) *Line 408: How is wind drag coefficient computed, and how did the authors make sure all simulations are consistent with this regard? i.e. is the wind contribution similar when waves are accounted for?*

The wind drag coefficient has been computed according to the formulation of Smith (1988), which estimates the coefficients for sea surface wind stress as a function of wind speed while accounting for a constant Charnock coefficient. These details are now specified in the revised manuscript.   In the discussion of the results, our aim was to methodically isolate the distinct contributions stemming from various forcing factors. To achieve this, we conducted a series of simulations in which we incrementally introduced different metocean forcings while ensuring that their effects remained as isolated as possible. As part of this approach, we deliberately did not consider the influence of waves on winds through adjustments to the Charnock coefficient. In other words, we kept the wind contribution consistent, irrespective of whether waves were

factored into the model. This deliberate choice allowed us to scrutinise and quantify the impact of other variables while minimising the confounding effects of altered wind conditions due to wave effects.

30) *Lines 413-414: What about the impact of the mixing length and its parameterisation (end of Section 3.3.5) on the current magnitude at the top of the water column?*

We did not perform sensitivity tests on the impact of the mixing length parameterisation employed. In Section 3.3.5 we introduced the turbulence model to highlight the difference in its choice with respect to former versions of ROMS, which include some of the additional spectral wave terms discussed in our study while using a different turbulence model.

31) *Line 433: I rather see a 1 and 3 cm/s increase depending on the configuration.*

The Reviewer is right. We have modified the text accordingly.

32) *Line 435: "[. . . ] improves the accuracy of the results". Given the results, this is all relative, i.e. whether taken at the surface or bottom. What are the rmsd for depth-averaged values and how do they compare?*

The results using the new spectral field exchange show an improved matching with the measurements. However, the improvements are visible when analysing the vertical profiles at specific time instants. Differences between the standard and full coupling seems very little when looking only at the time-series of the depth-averaged velocity computed by the 2WC and the new 2WF simulations, with respectively RMSE of 7.0592 and 7.0865 and RSQRT of 0.9761 and 0.9759. For this reason and since the entire paper is focused on the 3D modelling of wave-current interactions, we did not include the stats of depth-averaged currents in the paper.

33) *Line 437: Why jumping over 3 Figures? Also, this specific graph shows velocities (m/s) and heights (m) over the same axis, a different one should be used.*

The reference has been changed to Figure 12 in the track-change version of the manuscript which is the next Figure presented in the text. Old figure 17 has been modified according to the Reviewer's comment.

34) *Line 447: Given that the boundary remains close to the measurements location, too much uncertainty exists on whether or not the model develops a realistic 3D profile (i.e. taking into account the response to winds etc). In this context, the statement is speculative and should be investigated further.*

We removed this sentence from the text.

35) *Line 443-444: What do the authors mean here? Please develop.*

Probably there is a typo in the line numbers. At these lines, we just observed how wind aligned with tidal propagation accelerates further the free-surface velocity.

36) *Line 451-452: So why choosing this limited experimental data set or focusing on the upper section of the water column?*

To understand why we focus on this particular data set, please see the response to the third major

point. It's crucial to emphasise that the primary focus of our paper is to evaluate the added-value of incorporating additional spectral wave terms in the computation of wave-current interactions, rather than concentrating on the details of turbulence modelling in these interactions. We employed a standard turbulence model with standard boundary conditions to ensure a fair and consistent comparison between simulations with and without the spectral terms.

37) *Line 465: Not supported. Using more carefully designed numerical experiments, including synthetic cases would be extremely useful here.*

The sentence has been removed from the text.

38) *Line 472-473: Here, I think it is misleading to compare the wave breaking terms for bulk and full spectral quantities. The bulk representation has so far only been derived and used for the case of depth-induced wave breaking (Uchiyama et al., 2010; Kumar et al., 2012) and not for whitecapping, which is the dominant dissipative process for waves in the present case. Whitecapping affects mostly high-frequency components, so the comparison is not relevant in my opinion. This made me realise that the authors did not describe the full spectral representation of the wave forces (not that of dissipation).*

We agree with the Reviewer that directly comparing these terms may not be suitable, given that the bulk representation has primarily been applied in the context of depth-induced wave breaking rather than whitecapping, which is the dominant dissipative process for waves in our study area. However, we would like to clarify that our intention was not to compare bulk and full spectral representations of wave breaking. Instead, our objective was to assess the improvement resulting from our model developments, which include the representation of whitecapping dissipation in addition to depth-induced breaking. Since no depth-induced breaking occurs in the measuring location where the data discussed in this study have been collected, the observed changes in the solution are primarily due to the consideration of whitecapping effects. To make this point clearer, we have revised the corresponding text to better express the purpose of this comparison.

39) *Line 490: This is all relative, we speak of 0.5 cm/s differences while wave model performances are poor and RMSE for currents is 10 cm/s, with much larger errors locally.*

Here we were comparing the impact of the different spectral terms added, and so USS gives 0.5-0.8 cm/s out of a total difference between 2WC and 2WF of around 4-5 cm/s, which is not neglibible. Please also consider that the disparities in current magnitude are, in certain instances, greater than 3 cm/s. Furthermore, when addressing variations slightly below 1 cm/s in a Stokes drift that fluctuates between 10 and 5 cm/s, this accounts for approximately 10% of the overall value.

40) *Fig. 17: Please clarify the representation of water levels as it is not clear. Also, in some Figures, the vertical datum for depth is not clear.*

Figure 17 has been changed to clarify the representation of water levels.

41) *Line 497: Is that so? Unless there are inconsistencies in ROMS/CROCO, the peak*

*wavenumber used in the monochromatic formulation should be Doppler-shifted too in full-coupled configurations.*

The Reviewer is right. We have removed the sentence from the manuscript. Initially, this sentence aimed to elucidate the distinctions between one-way and two-way coupled simulation outcomes. However, we later decided to exclude the one-way simulations from the paper to enhance conciseness.

42) *Line 505: Please be consistent. In some sections of the manuscript, refraction processes are well represented and in this one, they are likely misrepresented. In both cases, the analysis is poorly supported by results.*

Since we have removed from the text the parts discussing wave processes in the surf zone, the depth-induced refraction process will be described only as likely misrepresented. We will deal with this issue using higher resolution nested models in shallow waters in our future modelling works.

43) *Line 506: So why not increasing the spatial resolution of the model? And why analysing processes such as depth-induced refraction and breaking?*

Please, see the response to the third major comment.

44) *Line 515-520: How is the surf zone identified here? Does that include region where whitecapping occurs? Like many sections and physical interpretations of results, this discussion is poorly supported.*

We have modified the text replacing the use of the term 'surf zone' to avoid any confusion.

45) *Line 544-545: I struggle to understand and believe the key message here.*

The observed slight phase shift in tidal floods between modelled and measured currents does not depend on random metocean forcing. As it is not related with wind and wave forcing, it does not impede our ability to assess the effects of wind and wave forcing on macro-tidal currents, which is the primary focus of our study. Thus, this phase shift is not a hindrance to the evaluation of our model developments that are primarily related to wave and wind influences on currents. A sentence has been added to the text to clarify these aspects.

[Figure]

Figure 1: **New model results in the extended computational domain**. Depth-averaged current magnitudes (coloured shading) with superimposed current vectors (black arrows) computed from the new simulation across its whole domain at the peak of the flood tidal flow the 2 August 2008 at 11 pm. White lines are bathymetric contours while black circles are the extracting (ADCP and AQDP) locations.

[Figure]

Figure 2: **New model results in the previous computational domain**. Depth-averaged current magnitudes (coloured shading) with superimposed current vectors (black arrows) computed from the new simulation across the old domain at the peak of the flood tidal flow the 2 August 2008 at 11 pm. White lines are bathymetric contours while black circles are the extracting (ADCP and AQDP) locations.

[Figure]

Figure 3: **Time series of newly computed tidal currents versus AQDP data**. Model results (lines) are compared with AQDP (dots) measurements in terms of (top) current magnitude and (bottom) 'going to' direction at 1 m above bottom (mab).

[Figure]

Figure 4: **Vertical profiles of newly computed tidal currents versus ADCP data**. Measured (squares) and modelled (continuous lines) vertical profiles of the current on 2 August 2008 at 2.30 pm during tidal reversal (left panel). Measured (squares) and modelled (continuous lines) vertical profiles of the current on 2 August 2008 at 11.00 pm during tidal ebb (right panel).

**Reply to Reviewer 2**

**General comment**

*This is an excellent manuscript. In this work, the authors discussed the implementation of WW3 and CROCO. They tested the coupled model using a representative case near the Bay of Somme and compared the simulation results against available observations. The paper is very well written. The experiments are designed appropriately. The equations used in the model look correct to me, but I did not derive the equations.*

We would like to thank the Reviewer for her/his positive feedback on our manuscript. We appreciate the Reviewer's thorough review and are pleased to hear that she/he found the work well-written and the experiments appropriately designed. In the following, we address both major and minor points raised.

**Major points**

1) *My major concern is, the simulated wave variables from the experiments are not different by a lot considering the feedback of the current velocity (Fig. 3, 7, 8, 9). Sometimes the new implementations are worse compared with the previous versions. Does it mean the fully coupled model has larger errors?*

Thank you for the feedback regarding the concern about the performance of the newly implemented version of the model. It's important to note that results in Figure 3 show that the predicted bulk wave parameters from the fully coupled model are very similar to those of the previous version of the coupled model, with comparable RMSE and R2 values. This means that the impact of spectral quantities on currents and water levels has very little impact on the effect of surface currents on waves, which is quite normal as the difference in surface current is max 4 cm/s for a current of the order of 25-30 cm/s (see Figure 10 and 11 of the track-change version of the manuscript). However, as for Figures 7, 8, and 9, these results are only from CROCOv1.1 version of the model. We have deliberately chosen not to include results from the newly implemented version in these particular figures, as our focus here was to compare the effects of the different forcing employed.

2) *It would also be desirable if the authors can discuss the extra computational cost introduced by the new implementations.*

The extra computational cost of including the additional spectral term exchanges in the fully coupled version is negligible in the present model configuration, accounting for less than 1% of the computational time required by the previous version of the coupled model. This is mainly because the spectral terms are in all cases calculated by the wave model. If we would made a comparison using WKB instead of WW3, we would have a much higher cost. It's not the

inclusion of the terms in CROCO that can add calculation time. However, it is important to note that the impact of these additional exchanges on computational cost may vary in larger domain configurations. We will include this information on the extra computational costs in the manuscript.

**Minor points**

1) *Line 79: why are the exceptional tides not simulated in this case?*

The exceptional tides were not simulated due to the unavailability of relevant data for that timeframe (september and march).

2) *Line 93: is the CROCO model adequate to resolve the small dunes? I think the small ripples/dunes can be as small as a few centimeters ( 150 d50).*

The current configuration of the CROCO model with a 100-meter horizontal resolution is not suitable for resolving small-scale features such as small dunes and ripples, which can be as small as a few centimeters in size. Our focus in this study is primarily on larger-scale processes, such as the interactions between currents and waves. However, we acknowledge the limitations in resolving fine sediment patterns. In future phases of our modelling efforts, we plan to enhance the model resolution through nesting, reaching a horizontal resolution of 10 meters and increasing the number of vertical layers. This approach will enable us to better represent tidal dunes, while smaller-scale wave ripples will be parameterised as bottom roughness.

3) *Lines 105-106: the text should be cleaned up.*

The text between lines 105-106 will be cleaned up. We acknowledge that the issue occurred during a minor revision phase when we were instructed to change only the title and availability statement. Unfortunately, an erroneous copy-paste during the final submission included internal notes from another manuscript. We apologise for this oversight.

4) *Line 125: it would be helpful if the author could add more about this equation. What is the unknown variable? How is the equation solved? How are $c_theta$ and $c_w$ represented in the equations?*

We have decided to replace the equation in question with the original one found in the WWIII manual to avoid any potential confusion. Any detailed information about its unknown variables and solution methods is available in the WWIII manual.

5) *Eq. (2): I think these are the equations actually solved in CROCO, why don't the authors put this part in Section 3.2?*

These equations are wave-averaged equations that CROCO solves specifically when wave-current interactions are considered. Following the Reviewer suggestion, we moved these equations to the previous section which introduces CROCO.

6) *Line 164: I think the monochromatic approximations are introduced in the latter sections. How about "... are computed from their monochromatic approximations, which are introduced in the latter sections"?*

The text will be modified as suggested to improve clarity.

7) *Line 309: the wind forcing resolution seems too coarse compared with the model resolution. Can the authors comment on this gap?*

We agree with the Reviewer's observation, and we will include a comment in the text to address this gap.

8) *Line 311: I am not sure about the meaning of mean grain size. Is it d50 (mass median diameter)?*

Yes, when we refer to 'mean grain size,' we are indeed referring to d50, which represents the mass median diameter of sediment particles. We will make this clarification in the text.

9) *Line 413: why does the flow fit better with observations?*

The improved fit with observations in the flow is attributed to the inclusion of waves propagating in the direction opposite to the wind stresses. This addition helps smooth out the overly sharp wind-induced current vertical profile at the free surface, a mechanism that is explained in detail in Groeneweg and Klopman (1998).

---

## Author Response (AR2)

**Reply to Reviewer's Second Revision**

**General comment:**

*Overall, I was not very satisfied with the authors' responses to my comments, as they were often lengthy, sometimes irrelevant or off the point. Some are highlighted below, where I either provide an answer to their response or further clarify my point.*

We appreciate the Reviewer's feedback and recognise the need for brevity and relevance in our responses. We have carefully reconsidered the points raised in this second revision and aim to address them more directly in this reply.

**Remaining major comments:**

1) *The authors argument to neglect the effect of waves on the wind drag and those from the wind and waves on the turbulence and vertical mixing at the surface remains unjustified. Since most parameterisations depend on bulk parameters such as the significant wave height, such effects can similarly be accounted for in both model configurations (monochromatic versus fully-spectral). These effects have been proven to have substantial effect on vertical current profiles, and should therefore be included given that the impact of the fully-spectral configuration is discussed with velocities taken near the surface or at the bottom. For instance, the role of whitecapping in accelerating currents near the surface cannot be discussed without accounting for the generation of turbulence by the same process, and the associated vertical mixing. As a result of the consistency claimed by the authors, some contributions might be biased.*

First, it should be noted that we have not neglected the impact of waves and winds on the turbulence and vertical mixing. These contributions are accounted for as described in subsections 3.3.4 and 3.3.5. The only thing that was neglected in our approach was the eventual spread in the drag coefficient associated to various sea states for a given wind speed. Indeed, for the ocean model wind stress computation, as we considered a constant Charnock coefficient, the drag coefficient was parameterized as a function of wind speed only, assuming that the sea state impact on the drag coefficient is only a function of the wind, and therefore neglecting an eventual variation of the drag for various sea states at a given wind speed.

To address the effects of considering wave-influenced wind drag on the computed vertical current profiles, we conducted a supplementary fully-spectral simulation. In this simulation, the wind stress is prescribed from that computed in the wave model (TWOX,TWOY in WW3), therefore accounting for the impact of varying sea states on the drag coefficient. The comparison of the current profiles predicted by this new simulation with those obtained using a wind stress computed with a constant Charnock parameter are shown in Figure 1 for various snapshots discussed in the paper. The impact is weak and will not affect the main outcome of our study. We have added a sentence mentioning this sensitivity experiment in the manuscript.

[Figure]

Figure 1: Vertical velocity profiles computed by the original (black line) and new simulation (red line) at the adcp location at time instants used to discuss the main findings of the study, namely 02/08/2008 14:30 (top-left panel, see Figure 11–C), 02/08/2008 18:00 (top-right panel, see Figure 10–C), 02/08/2008 23:00 (bottom-left panel, see Figure 11–D), 03/08/2008 17:30 (bottom-right panel, see Figure 10–D).

2) *The novelty claimed by the authors remains a novelty within the present modelling system (CROCO). This should be stressed more clearly in the manuscript. As opposed to what the authors respond, other modelling systems employing the proposed fully-spectral approach have been applied in regional macrotidal settings.*

In the revised version of the manuscript we have stresses more clearly that the novelty of our work is innovative within the CROCO modelling system.

**Comments on the authors' responses:**

Response to major comment 1): "While the references cited by the Reviewer describe developments akin to ours, all aimed at achieving a more comprehensive description of wave-current interactions, our study delves into aspects specific to the employed modelling system, setting it apart from the frameworks mentioned by the Reviewer in several key ways. Regarding technical aspects, our coupled model uses different hydrodynamic and wave spectral models than the cited systems. [...]" *I do not agree. All studies I mentioned already compute forcing terms from the full spectrum. This is completely independent from the modelling system, if the equations are the same. My point is: what the authors claim is a novelty in their study, is actually only a novelty within their modelling system since it already exists in others (e.g. SCHISM, MO-HID, not fully in ROMS and probably others). "Our work, in line with these recent advances, concentrates on its adaptation for intermediate-water macro-tidal conditions, distinguishing it from the cited modelling frameworks which are more nearshore-oriented." Have the authors actually read the references I provided or just cited them in response? Some actually consider macrotidal environments at the regional scale. A modelling system like SCHISM can nowadays actually be used at the global scale, thanks to the use of unstructured grids.*

In the revised version of the manuscript we have changed the text accordingly.

Response to specific comment 2): "We will explicitly narrow down the study objective from 'modelling wave-current interactions in coastal areas' to 'modelling interactions of macro-tidal currents with winds and waves'. This adjustment will ensure greater clarity and alignment with the primary goals of our research." *No, it does not bring more clarity: macro-tidal currents are also found in shallow water regions. What, I think, the authors mean is that their model configuration is at the "coastal" scale, in a macro-tidal environment and away from the nearshore breaking wave region (justifying to switch off this latter term).*

We agree with the Reviewer's comment and appreciate her/his guidance in enhancing the clarity of our work's objectives. In the revised manuscript we clarify that our main objective is to model macro-tidal currents affected by winds and waves within a coastal scale configuration, forced by tidal atlas along with wind and wave forcing, to investigate wave-current interactions in intermediate water depths, away from the nearshore breaking wave region. The approach will enables us to inform higher-resolution nested models in the nearshore. This refined objective more accurately reflects our research's scope and primary goals.

Response to major comment 3): "To provide a comprehensive context for our study and elucidate the rationale for our choice of modelling application, it is essential to place our research within the framework of its funding project." *I get that funding bodies expect specific deliverables, but how does that justify shortcomings in scientific articles? In my opinion, this answer has nothing to do in the response to the question: why is the selected site relevant to the overall aim of the paper?*

The selected site is relevant to the overall aim of the paper because it provides comprehensive measurements of macro-tidal currents across the water column, alongside free surface wave

data. These observations were crucial for validating our modelling, which aims to replicate tidal currents affected by winds and waves within a coastal scale configuration. Our study leverages these measurements to investigate wave-current interactions in intermediate water depths. This focus aligns closely with our objectives, enabling us to utilise the data to not only validate our model but also to understand the impacts of winds and waves on tidal currents in such environments.

Response to major comment 2): "These additional results convincingly demonstrate that the boundary was situated at a sufficient distance from the measurement location where vertical velocity profiles were examined." *This is quite of a shortcoming. Differences in terms of bottom velocities between new and old model configurations are of the order (if not greater) of the differences obtained with the monochromatic and full-spectral approaches. Thus, it completely seems relevant to keep the boundary away from the measurement site. Furthermore, it is more consistent with the regional scale the authors claim to aim. From the authors response, it is not clear whether or not they kept their original configuration or switch to this extended domain.*

It is important to place our answer in the context of the original comment from the Reviewer: "Considering the importance given to the vertical shear, how can we be sure that the boundary is taken sufficiently far away from the location where measurements were obtained and vertical velocity profiles discussed? I suspect that the boundary is too close to the measurement stations in order to discuss differences of the order of 1 cm/s at the top layer of the water column."

In response to the Reviewer's concern about the model boundary's proximity to measurement stations, our additional analyses compared the original simulation forced only by tide with an extended model configuration in terms of current profiles at the measuring station. These analyses revealed that differences in vertical shear between both configurations are minimal at those instants at which we extracted and discussed vertical velocity profiles, especially in the top layers of the water column, confirming the adequacy of our boundary placement. While a single vertical profile showed bottom velocity differences (Figure 2) of a similar magnitude to those observed when comparing monochromatic and fully-spectral runs with the original tide-only configuration, these differences diminish rapidly within the water column, becoming negligible near the surface. Also, these differences likely stem from the different interpolation of boundary conditions, both in terms of hydrodynamic forcing associated with the considered tidal constituents and modelled bathymetry, and can be appreciated only at tidal current reversal.

Overall, our analysis indicate that extending the model boundaries further does not improve the model prediction of the observed vertical shear pattern in the mid-water column shown by the ADCP data, which we believe is the main reason for the Reviewer's original comment. These findings suggest that our model configuration is sufficiently robust to discuss observed vertical shear at the top layer of the water column. Given the study's focus on efficiently predicting wave-current interactions by means of a coastal scale configuration forced with tidal atlas, expanding the model domain would increase computational costs without yielding substantial improvements in outcomes. Thus, our modelling approach balances accuracy with computational efficiency, aligning with the study's main objective.

[Figure]

Figure 2: **Vertical profiles of newly computed tidal currents versus ADCP data**. Measured (squares) and modelled (continuous lines) current vertical profiles on 2 August 2008 at 14:30 pm during tidal reversal.

Response to specific comment 12): *My point was that waves generated within a 400 km fetch are not usually considered swell.*
The text has been corrected accordingly.

Response to specific comment 21): *I do not understand the authors' answer. My question, which is extremely relevant for the present work: what is the formulation used for the wind drag coefficient? How is the wind surface stress computed? Short waves as those found in the English Channel will have a strong impact on those quantities. Consequently, the authors choice will affect the wind contribution to mixing and the vertical velocity profiles.*
Please see the response to the first major point.

Response to specific comment 29): *I understand the logic, but I think it is wrong for two reasons: 1) it first means that the wind contribution to changes in vertical velocity profiles is overestimated; 2) the wind stress will not change with the choice of the coupling approach (monochromatic vs fully-spectral), which means that the authors can still perform consistent comparisons, but with better estimates of the effects of waves on the wind stress (and not just "the effect of waves on wind" as written in the revised manuscript).*
Please see the response to the first major point.

Response to specific comment 32): "Also, since the entire paper is focused on the 3D modelling of wave-current interactions, we do not see the point in computing stats associate to

depth-averaged values." *Then please discuss the sensitivity of current vertical profiles to turbulence (both TKE and mixing length), including that produced by waves and wind. These will have a major impact on the value at specific height along the water column. For instance, how does the choice for the mixing length parameterisation impact the value of surface currents: does it vary more (or less) compared to the differences between monochromatic and fully-spectral configurations?*

We agree that the interaction between turbulence, wind- and wave-induced mixing is crucial for accurately modelling wave-current interactions. We have indeed accounted for turbulence generated by winds and waves, as detailed and discussed in various part of the manuscript (see response to major point 1). These contributions are treated consistently across both the monochromatic and fully-spectral model configurations, with a focus on discussing the differences between these approaches. While our study does provide a foundation for future work in this area, including a detailed examination of turbulence modelling would require a dedicated sensitivity analysis, which is out of the scope of the present study.

Response to specific comment 36): "It's crucial to emphasise that the primary focus of our paper is to evaluate the added-value of incorporating additional spectral wave terms in the computation of wave-current interactions, rather than concentrating on the details of turbulence modelling in these interactions. We employed a standard turbulence model with standard boundary conditions to ensure a fair and consistent comparison between simulations with and without the spectral terms." *Similar to the waves effect on the wind stress, the two configurations can still be compared with standard practices when dealing with turbulence production and dissipation at the surface within the coastal region. That is including the contribution of winds and short waves in the production of TKE near the surface, which will have a major role in mixing and hence vertical velocity profiles. Again, this is because the two configurations tested here (monochromatic or fully-spectral) will not affect the source of TKE from waves.*

Please see the above response to the previous Reviewer's comment and the that to the first major point.

**Specific comments on revised manuscript:**

1) *Line 83: "the use of...".*

The text has been correct.

2) *Line 117-118: "This indicates a not fully developed wind sea with energetic swells that develop over 400 km-long fetches.". Please adjust (cf. my comment above).*

The text has been correct.

3) *Some references cited in the text are missing in the revised version of the manuscript.*

The references have been updated.